# When and How Unlabeled Data Provably Improve In-Context Learning

**Yingcong Li**[1,4]     **Xiangyu Chang**[2]     **Muti Kara**[3]
**Xiaofeng Liu**[1]     **Amit Roy-Chowdhury**[2]     **Samet Oymak**[1]

[1]University of Michigan     [2]University of California, Riverside     [3]Bilkent University     [4]NJIT

## Abstract

Recent research shows that in-context learning (ICL) can be effective even when demonstrations have missing or incorrect labels. To shed light on this capability, we examine a canonical setting where the demonstrations are drawn according to a binary Gaussian mixture model (GMM) and a certain fraction of the demonstrations have missing labels. We provide a comprehensive theoretical study to show that: (1) The loss landscape of one-layer linear attention models recover the optimal fully-supervised estimator but completely fail to exploit unlabeled data; (2) In contrast, multilayer or looped transformers can effectively leverage unlabeled data by implicitly constructing estimators of the form $\sum_{i \geq 0} a_i (X^\top X)^i X^\top y$ with $X$ and $y$ denoting features and partially-observed labels (with missing entries set to zero). We characterize the class of polynomials that can be expressed as a function of depth and draw connections to Expectation Maximization, an iterative pseudo-labeling algorithm commonly used in semi-supervised learning. Importantly, the leading polynomial power is exponential in depth, so mild amount of depth/looping suffices. As an application of theory, we propose looping off-the-shelf tabular foundation models to enhance their semi-supervision capabilities. Extensive evaluations on real-world datasets show that our method significantly improves the semisupervised tabular learning performance over the standard single pass inference.

## 1 Introduction

In-context learning (ICL) is an intriguing capability of modern language models and has enjoyed remarkable empirical success (Brown et al., 2020; Min et al., 2022). This success is also being extended to multimodal scenarios (Zhou et al., 2024) as well as other modalities such as tabular data (Hollmann et al., 2022). The push toward test-time scaling and long-context models (Snell et al., 2024; Guo et al., 2025) has further boosted the benefits of ICL by allowing the model to ingest a large number of demonstrations. For instance, in "Many-shot in-context learning" paper, Agarwal et al. (2024) demonstrate that pushing more examples into context window can substantially boost the accuracy. MAPLE (Chen et al., 2025) improves many-shot ICL by pseudo-labeling high-impact unlabeled examples and incorporating them into the prompt. The many-shot ICL setting naturally raises the question of when and how ICL can succeed with weaker supervision. As we can harness longer context models to boost predictive accuracy, we may indeed run out of high-quality demonstrations with verified answers/chain-of-thoughts and may want to utilize weaker data sources. This motivates our central question:

> **Q: When and how can transformers learn in context from unlabeled data?**

We primarily investigate this question under a semisupervised ICL (SS-ICL) setting with Gaussian mixture models (GMMs). Formally, given a prompt containing a dataset of feature-label pairs $(x_i, y_i)_{i=1}^n \in \mathbb{R}^d \times \mathbb{R}$ as demonstrations and a query feature $x$ (see Eq. (3)), a model trained for ICL

learns to predict the corresponding output $y$ given prompt. For ICL with a supervised binary GMM model, we have $\boldsymbol{x}_i \sim \mathcal{N}(\boldsymbol{\mu}_{y_i}, \sigma^2 \boldsymbol{I})$ and $y_i \in \{-1, 1\}$, $i \in [n]$, and the component means $\boldsymbol{\mu}_{\pm 1}$ that parameterize the classification task are sampled from a prior task distribution. This prompt model is well studied under various fully-supervised settings (Garg et al., 2022; Von Oswald et al., 2023; Ahn et al., 2023; Akyürek et al., 2023; Mahankali et al., 2024; Collins et al., 2024; Shen et al., 2024) where each demonstration includes a clearly labeled output. In our SS-ICL setting, only $m$ out of $n$ total samples have correct labels ($m \leq n$) either $-1$ or $1$, and remaining labels are unknown and fed to the model as $y_i = 0$.

In this work, we provide a comprehensive theoretical and empirical study of attention models with varying depths when trained with SS-ICL. Our analysis reveals the importance of *depth*: Despite being able to implement the optimal fully-supervised estimator, single-layer linear attention completely fails to leverage unlabeled examples. In contrast, deeper or looped transformer architectures can emulate strong semi-supervision algorithms, approaching the performance of the Bayes-optimal classifier as depth increases. Informed by the importance of depth/looping, we also devise semisupervision strategies for tabular foundation models. Our specific contributions are:

⋄ **Landscape of one-layer linear attention (§3):** We study the optimization landscape of single-layer linear attention for the SS-ICL problem under an isotropic task prior. We prove that the global minimum of the loss function returns the plug-in estimator (see Eq. (SPI)), i.e., $\hat{y} = \text{sgn}(\boldsymbol{x}^\top \hat{\boldsymbol{\mu}})$ with $\hat{\boldsymbol{\mu}} = \boldsymbol{X}^\top \boldsymbol{y}$, where $\boldsymbol{X} \in \mathbb{R}^{n \times d}$ represents features and $\boldsymbol{y} \in \mathbb{R}^n$ denotes partially-observed labels (with missing entries set to zero) of the ICL demonstrations. This implies that 1-layer model learns Bayes-optimal classifier in the fully-supervised setting, but completely fails to make use of unlabeled data.

⋄ **Depth is crucial but shallow can suffice (§4):** We show that multilayer linear attention can emulate semisupervised learners by implementing polynomial estimators of the form

$$\hat{\boldsymbol{\mu}} = \sum_{i=0}^{K} a_i (\boldsymbol{X}^\top \boldsymbol{X})^i \boldsymbol{X}^\top \boldsymbol{y}. \tag{1}$$

Crucially, an $L$-layer (or looped) attention can express up to $K = O(3^L)$ powers, highlighting that logarithmic depth suffices to represent high-degree monomials. We provide characterizations of the set of expressible polynomials through different constructions (where each layer gets to update the features or labels of the previous layer). Corroborating these, experiments reveal that shallow models with $L \geq 2$ already achieve strong results and their performance can be approximately predicted through an eigen-estimator combining $i = 0$ and $\infty$ (see (SSPI-$k$)).

⋄ **What learner attention emulates?** In Section 4.3, we describe how each attention block can update the label estimates by emulating expectation-maximization (for linear attention) or belief propagation (for softmax attention). For instance (1) can be interpreted as the model implicitly conducting an *Expectation-Maximization* algorithm: Starting with the supervised estimator $\hat{\boldsymbol{\mu}}_0 = \boldsymbol{X}^\top \boldsymbol{y}$, each term $(\boldsymbol{X}^\top \boldsymbol{X})^i \boldsymbol{X}^\top \boldsymbol{y}$ can be viewed as a sequence of pseudo-labeling (expectation) $\hat{\boldsymbol{y}}_i = \boldsymbol{X} \hat{\boldsymbol{\mu}}_{i-1}$ and training (maximization) $\hat{\boldsymbol{\mu}}_i = \boldsymbol{X}^\top \hat{\boldsymbol{y}}_i$ steps. Corroborating this, we show that softmax-attention and softmax-transformer models similarly benefit from increasing depth and can emulate semisupervised learners competitive with Bayes limit (see Fig. 2c).

⋄ **Applications to Tabular FMs (§5):** Tabular foundation models such as TabPFN (Hollmann et al., 2022, 2025), TabICL (Qu et al., 2025) and TabDPT (Ma et al., 2025) represent a suitable application of theory as they also model the ICL examples with a single token. To harness unlabeled examples, we propose a novel strategy that iteratively creates soft pseudo-labels by *explicitly looping the tabular FM* while controlling validation risk. Focusing on the few-shot learning setting where TabPFN-v2 (Hollmann et al., 2025) excels, we demonstrate that our approach can significantly improve predictive performance on various real-world datasets.

## 1.1 Related Work

**Theoretical Analysis of In-Context Learning** Recent work has developed theoretical frameworks for understanding in-context learning in transformers. Akyürek et al. (2023), Von Oswald et al. (2023) and Dai et al. (2023) demonstrated that transformers emulate gradient descent during ICL. Xie et al. (2022) offered a Bayesian perspective, while Zhang et al. (2024) showed transformers learn linear models in-context. Ahn et al. (2023) established they implement preconditioned gradient

descent, and Mahankali et al. (2024) proved one-step gradient descent is optimal for single-layer linear attention. Multiple works (Li et al., 2023; Yang et al., 2024; Li et al., 2024; Bai et al., 2023; Shen et al., 2024) studied the generalization capability of transformers. However, these exclusively focus on fully-supervised settings, leaving a critical gap in understanding how transformers handle partially labeled data—a common real-world scenario. Our work addresses this gap by providing the first theoretical characterization of semi-supervised in-context learning. Wang et al. (2024) considers a setting where the model observes demonstrations of the form (query, response$_i$, reward$_i$) and aims to correct its response based on the reward sequence. Our work has a different focus as it highlights that the model can correct/impute the missing labels using implicit feedback from labeled demonstrations.

**Semi-Supervised Learning**    Traditional semi-supervised learning (SSL) aims to leverage unlabeled data to improve classifier performance. For linear classifiers, Oymak & Gulcu (2021) character-ized self-training iterations and demonstrated rejecting low-confidence samples; further theoretical analyses of self-training/pseudo-labeling cover deep networks (Wei et al., 2020). For Gaussian Mixture Models (GMMs), Lelarge & Miolane (2019) quantified maximal improvement from un-labeled data, while Krishnapuram et al. (2004) developed graph-based priors. Learning GMMs via Expectation-Maximization (EM) or pseudo-labeling, especially with few labels, is well-studied. Ratsaby & Venkatesh (1995) provided early PAC-style bounds for GMMs learned from few labeled and many unlabeled points. Balakrishnan et al. (2017) offered further statistical guarantees for EM. Nigam et al. (2000) demonstrated empirically that EM (viewable as iterative pseudo-labeling Xu et al. (2024)) with pseudo-labels significantly reduces text classification error using unlabeled documents. These foundational works, with ongoing research in areas like agnostic learning (Kwon & Caramanis, 2020) underpin many SSL concepts. While these works established fundamental principles, they did not consider how these concepts apply to in-context learning with transformers. A most recent concurrent work (Liu & Yang, 2026) makes a similar observation to ours, showing that softmax attention approximates an EM estimator in a sem-supervised ICL setting, but with a different focus on the underlying model and data regime. Our contribution bridges this gap by showing how transformer depth enables effective utilization of unlabeled examples within the prompt, essentially implementing semi-supervised learning without parameter updates.

## 2  Problem Setup and Preliminaries

We study ICL in the setting of semi-supervised classification, where the in-context demonstrations are drawn from a binary Gaussian mixture model (GMM). We begin by introducing the following core notation: Denote the set $\{1, 2, \cdots, n\}$ as $[n]$ and use bold letters, such as $\boldsymbol{x}$ and $\boldsymbol{X}$, to represent vectors and matrices, respectively. Let $Q(\cdot)$ function return the right tail of the standard normal distribution. We use sgn$(\cdot)$ denote the sign function which is defined as follows: $\text{sgn}(x) = \begin{cases} 1, & x \geq 0 \\ -1, & x < 0 \end{cases}$.

### 2.1  Semi-supervised Data Model

Consider a $d$-dimensional semi-supervised binary GMM with $n$ examples $(\boldsymbol{x}_i, y_i)_{i=1}^{n}$, where $\boldsymbol{x}_i \in \mathbb{R}^d$ denotes the feature vector and $y_i \in \{-1, 0, 1\}$ represents the corresponding observed label, with $y_i = 0$ indicating a missing label, and each label is revealed independently with probability $p \in [0, 1]$. Specifically, the data is generated as follows (for each $i \in [n]$):

$$\boldsymbol{x}_i = y_i^c \cdot \boldsymbol{\mu} + \boldsymbol{\xi}_i \quad , \quad y_i = \begin{cases} y_i^c, & \text{w.p.} \quad p \\ 0, & \text{w.p.} \quad 1-p \end{cases} \quad \text{and} \quad y_i^c = \begin{cases} 1, & \text{w.p.} \quad 1/2 \\ -1, & \text{w.p.} \quad 1/2 \end{cases}. \tag{2}$$

Here $\boldsymbol{\mu} \sim \text{Unif}(\mathbb{S}^{d-1})$ denotes the task mean, which is sampled uniformly from the unit sphere, and $\boldsymbol{\xi}_i \sim \mathcal{N}(0, \sigma^2 \boldsymbol{I})$ is the random noise with $\sigma \geq 0$ being the noise level that controls the variability of $\boldsymbol{x}_i$ around its mean. $y_i^c$ denotes the true class label that is uniform over $\{-1, 1\}$. Observe that $p = 1$ corresponds to fully-supervised learning and $p = 0$ corresponds to fully-unsupervised learning.

### 2.2  In-context Learning and Linear Attention

We build on the setting of (Garg et al., 2022; Mahankali et al., 2024; Zhang et al., 2024; Li et al., 2024) and construct the in-context prompts with examples drawn from the model (2) as follows.

**Prompt Generation** Given a task vector $\boldsymbol{\mu} \sim \text{Unif}(\mathbb{S}^{d-1})$, we sample $(n + 1)$ in-context demonstrations $(\boldsymbol{x}_i, y_i)_{i=1}^{n+1}$ according to (2) and construct the prompt

$$\boldsymbol{Z} = \begin{bmatrix} \boldsymbol{x}_1 & \boldsymbol{x}_2 & \cdots & \boldsymbol{x}_n & \boldsymbol{x} \\ y_1 & y_2 & \cdots & y_n & 0 \end{bmatrix}^{\top} \in \mathbb{R}^{(n+1)\times(d+1)}. \tag{3}$$

We will investigate training a transformer such that given $\boldsymbol{Z}$ as prompt, it correctly predicts the label $y := y_{n+1}^c$ of the query $\boldsymbol{x} := \boldsymbol{x}_{n+1}$ through ICL.

**Model Architecture** Our work primarily focuses on training of linear attention models. Given any prompt $\boldsymbol{Z} \in \mathbb{R}^{(n+1)\times(d+1)}$, which can be treated as a sequence of $(d + 1)$-dimensional tokens, the linear attention mechanism outputs

$$\text{att}(\boldsymbol{Z}; \mathcal{W}) = (\boldsymbol{Z}\boldsymbol{W}_q\boldsymbol{W}_k^{\top}\boldsymbol{Z}^{\top})\boldsymbol{M}\boldsymbol{Z}\boldsymbol{W}_v \tag{4}$$

where $\mathcal{W} := \{\boldsymbol{W}_k, \boldsymbol{W}_q, \boldsymbol{W}_v \in \mathbb{R}^{(d+1)\times(d+1)}\}$ denotes the set of the key, query and value weight matrices. Therefore, given the prompt matrix $\boldsymbol{Z} \in \mathbb{R}^{(n+1)\times(d+1)}$ as input, the attention mechanism outputs a $(n + 1)$-length sequence (i.e., $\text{att}(\boldsymbol{Z}; \mathcal{W}) \in \mathbb{R}^{(n+1)\times(d+1)}$). Note that the label for the query $\boldsymbol{x}$ is excluded from the prompt $\boldsymbol{Z}$. Similar to Ahn et al. (2023), we consider a training objective with a mask $\boldsymbol{M} = \begin{bmatrix} \boldsymbol{I}_n & 0 \\ 0 & 0 \end{bmatrix}$ to prevent input tokens from attending to the queries. To ensure that all in-context examples are treated equally and that the model remains invariant to their order/position, we do not apply a causal mask following Ahn et al. (2023). In contrast, Li et al. (2025) explores the use of causal masking in multi-layer linear attention and analyzes its impact on the final prediction.

Building upon the single-layer linear attention mechanism of (4), we can extend our model to multiple layers to capture more complex patterns. Consider optimizing an $L$-layer linear attention model and let $\boldsymbol{Z}_\ell$ be the input of $\ell$th layer, $\ell \in [L]$. Additionally, let $\mathcal{W}_\ell := \{\boldsymbol{W}_{k\ell}, \boldsymbol{W}_{q\ell}, \boldsymbol{W}_{v\ell} \in \mathbb{R}^{(d+1)\times(d+1)}\}$ be the corresponding weight matrices of $\ell$th layer. Then, recalling the attention mechanism (4), the input prompt of $\ell$th layer is defined by

$$\boldsymbol{Z}_\ell = \boldsymbol{Z}_{\ell-1} + \text{att}(\boldsymbol{Z}_{\ell-1}; \mathcal{W}_{\ell-1}) \qquad \text{for} \qquad \ell = 2, \ldots L, \tag{5}$$

and $\boldsymbol{Z}_1 = \boldsymbol{Z}$. We focus on the next-token prediction setting, where the model makes a prediction based on the final query token $[\boldsymbol{x}^{\top} \ 0]^{\top}$. Let $\boldsymbol{h} \in \mathbb{R}^{d+1}$ denote the linear prediction head. We define the output of the $L$-layer linear attention model at the last (query) token as

$$f_{\text{att-}L}(\boldsymbol{Z}) = \boldsymbol{h}^{\top}\text{att}(\boldsymbol{Z}_L; \mathcal{W}_L)_{[n+1]}. \tag{6}$$

Recalling the sign function, the predicted label for $\boldsymbol{x}$ is given by $y_{\text{att-}L}(\boldsymbol{Z}) = \text{sgn}(f_{\text{att-}L}(\boldsymbol{Z}))$.

**Model Training** With our attention-based architecture established, we now turn to the training procedure and evaluation metrics. Consider the ICL setting where each input prompt $\boldsymbol{Z}$ (cf. (3)) corresponds to a randomly sampled task vector $\boldsymbol{\mu} \sim \text{Unif}(\mathbb{S}^{d-1})$ and let $\ell(\cdot) : \mathbb{R} \to \mathbb{R}$ be the loss function. Additionally, define the set of attention weights $\mathcal{W}^{(L)} := \cup_{\ell=1}^{L} \mathcal{W}_\ell \in (\mathbb{R}^{(d+1)\times(d+1)})^{3L}$. The objective of $L$-layer linear attention takes the following form:

$$\min_{\mathcal{W}^{(L)}, \boldsymbol{h}} \mathcal{L}_{\text{att-}L}(\mathcal{W}^{(L)}, \boldsymbol{h}) \qquad \text{where} \qquad \mathcal{L}_{\text{att-}L}(\mathcal{W}^{(L)}, \boldsymbol{h}) = \mathbb{E}\left[\ell(y, f_{\text{att-}L}(\boldsymbol{Z}))\right]. \tag{7}$$

Here, $y = y_{n+1}^c$ and the expectation subsumes the randomness of $\boldsymbol{\mu}$ and $(\boldsymbol{\xi}_i, y_i)_{i=1}^{n+1}$. The search space for $\mathcal{W}^{(L)}$ is $(\mathbb{R}^{(d+1)\times(d+1)})^{3L}$, and for $\boldsymbol{h}$ is $\mathbb{R}^{d+1}$.

## 3 Loss Landscape of One-layer Linear Attention under SS-ICL

Previous work (Ahn et al., 2023; Li et al., 2024; Mahankali et al., 2024) has shown that an optimized single-layer linear attention implements a form of preconditioned gradient descent over the linear in-context demonstrations provided within the prompt. However, to the best of our knowledge, prior studies have not addressed the semi-supervised setting, where some in-context labels are missing. In this section, we analyze the optimization behavior of single-layer linear attention under the semi-supervised binary GMM setting described in Section 2, and demonstrate that the single-layer model learns the optimal fully-supervised learner, but fails to utilize the unlabeled data.

We begin with the following optimal supervised label estimator under our problem setting.

**Supervised Plug-in (SPI) Estimator** The plug-in method is a classical approach for supervised classification problems, aiming to find a linear combination of features that separates different categories. Under our problem setting, it also serves as the asymptotically Bayes-optimal estimator given only labeled data (Hastie et al., 2009; Devroye et al., 2013). Consider the binary semi-supervised GMM problem described in (2) with dataset $(\boldsymbol{x}_i, y_i)_{i=1}^n$, and let $\mathcal{I} \subset [n]$ represent the indices of labeled samples, e.g., $y_i \neq 0$ for $i \in \mathcal{I}$. The SPI estimator returns the task mean

$$\hat{\boldsymbol{\mu}}_s = \frac{1}{|\mathcal{I}|} \sum_{i \in \mathcal{I}} y_i \boldsymbol{x}_i. \tag{SPI}$$

We next present the following theorem establishes that, under isotropic task prior, optimal single-layer linear attention is equivalent to the SPI estimation.

**Theorem 1** *Let the prompt (cf. (3)) be generated as described in Section 2.2. Consider the objective (cf. (7)) with $L = 1$ and squared loss function $\ell(y, \hat{y}) = (y - \hat{y})^2$, and denote the optimal prediction as $y_{att\text{-}1}^{\star}(\boldsymbol{Z})$. Let $\hat{\boldsymbol{\mu}}_s$ represent the SPI estimator defined in (SPI). Then, for any $\boldsymbol{Z}$ from (3), we have*

$$y_{att\text{-}1}^{\star}(\boldsymbol{Z}) = sgn(\boldsymbol{x}^{\top}\hat{\boldsymbol{\mu}}_s). \tag{8}$$

*Additionally, its classification error obeys*

$$\mathbb{P}(y_{att\text{-}1}^{\star}(\boldsymbol{Z}) \neq y) = \mathbb{E}_{g \sim \mathcal{N}(0,1), h \sim \mathcal{X}_{d-1}^2} \left[ Q\left( \frac{1 + \varepsilon_\sigma g}{\sigma \sqrt{(1 + \varepsilon_\sigma g)^2 + \varepsilon_\sigma^2 h}} \right) \right] \tag{9}$$

$$\leq Q\left( \frac{1 - 10d\varepsilon_\sigma^2}{\sigma} \right) + e^{-d} + e^{-1/8\varepsilon_\sigma^2}$$

*where we define $\varepsilon_\sigma = \sigma / \sqrt{np}$ and $\mathcal{X}_d^2$ defines chi-squared distribution with d degrees of freedom.*

The proof of Theorem 1 is deferred to Appendix B. Eq. (8) shows that one-layer linear attention model indeed implements the optimal supervised predictor, assuming access to $np$ labeled examples. Therefore, the classification error corresponds exactly to that of the SPI estimator. The supervised classification problem has been extensively studied (Bartlett et al., 2006; Belkin et al., 2018; Montanari et al., 2019; Thrampoulidis et al., 2020; Chatterji & Long, 2021; Cao et al., 2021; Wang & Thrampoulidis, 2022; Deng et al., 2022), with most existing work focusing on a single classification task in asymptotic data or overparameterized regimes. In contrast, within the ICL framework considered in our setting, the task mean $\boldsymbol{\mu}$ is randomly sampled, and the classification error is computed by averaging over random draws of $\boldsymbol{Z}$, $y$, and $\boldsymbol{\mu}$. Accordingly, in (9), we express the error in a simplified form as an expectation.

The experimental results in Figure 1 support Theorem 1, where dark blue circular markers represent the performance of the single-layer linear attention model, blue curves show the classification accuracy of the SPI estimator, and the red dotted curves depict the accuracy $1 - \mathbb{P}(y_{att\text{-}1}^{\star}(\boldsymbol{Z}) \neq y)$ as computed from (9). The alignments of these curves empirically validate Theorem 1. Implementation details and further discussion are provided in Section 5. Based on these results, we reach the following conclusion:

*1-layer linear attention learns optimal supervised estimator but doesn't benefit from unlabeled data.*

As shown in Figs 1b and 1c, when the number of labeled samples ($np = 10$) is fixed, increasing the number of unlabeled examples (even up to $\sim 10000$) has no effect on performance, as the dark blue markers remain at the same level.

At first glance, this may seem counterintuitive—while the data is unlabeled, it still contains information about the classification feature. For instance, the mean of the data points carries relevant information, and one might expect the model to extract and leverage this for better predictions. This expectation is particularly reasonable when a large amount of unlabeled data is available, as the sample covariance matrix approximates the population covariance, i.e., $\mathbb{E}[\boldsymbol{X}^{\top}\boldsymbol{X}/n] = \boldsymbol{\mu}\boldsymbol{\mu}^{\top} + \sigma^2\boldsymbol{I}$ where $\boldsymbol{X} = [\boldsymbol{x}_1, \boldsymbol{x}_2, \cdots, \boldsymbol{x}_n]^{\top} \in \mathbb{R}^{n \times d}$. The key insight into why single-layer attention fails to leverage unlabeled data lies in the expectation structure. In our isotropic GMM setting where $\boldsymbol{\mu} \sim \text{Unif}(\mathbb{S}^{d-1})$, the sample covariance matrix converges to $\mathbb{E}[\boldsymbol{X}^{\top}\boldsymbol{X}/n] = \mathbb{E}[\boldsymbol{\mu}\boldsymbol{\mu}^{\top}] + \sigma^2\boldsymbol{I} = (1/d + \sigma^2)\boldsymbol{I}$, which contains no task-specific information. The expectation across multiple tasks loses the signal from $\boldsymbol{\mu}$. This

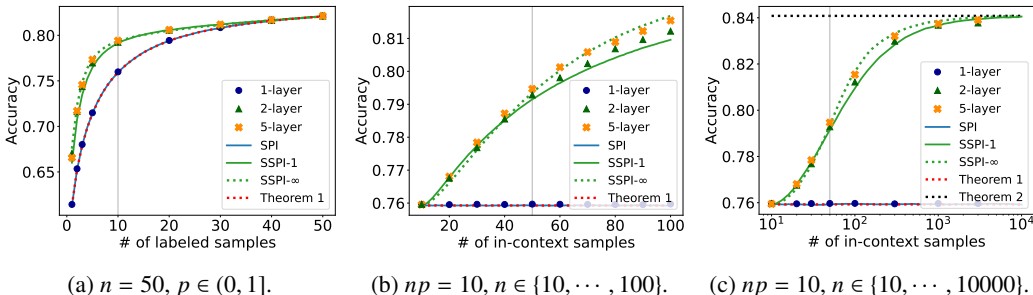

(a) $n = 50$, $p \in (0, 1]$.     (b) $np = 10$, $n \in \{10, \cdots, 100\}$.     (c) $np = 10$, $n \in \{10, \cdots, 10000\}$.

Figure 1: Experimental results support our theoretical findings presented in Sections 3 and 4. In all three subfigures, blue, green, and orange markers represent the results of 1-, 2-, and 5-layer linear attention models, respectively. The SPI estimator (cf. (SPI)), SSPI-1, and SSPI-∞ (cf. (SSPI-$k$)) are shown as blue solid, green solid, and green dotted curves, respectively. The red dotted curves in all subfigures correspond to the single-layer/SPI results described in Eq. (9) of Theorem 1, while the black dotted line in Fig. 1c corresponds to Eq. (13) of Theorem 2. Additional details and discussion can be found in Sections 3, 4, and 5.

explains why single-layer attention, operating in a meta-learning framework across many tasks rather than optimizing for a single fixed task, cannot extract useful information from unlabeled data.

In the following section, we study multi-layer linear attention and demonstrate that it has the ability to propagate $X^\top X$ into deeper layers, thereby enabling the model to utilize the unlabeled data.

# 4 Multi-layer Attention and the Benefits of Depth

In this section, we explore how deeper attention models can effectively utilize the unlabeled data. Let

$$X = \begin{bmatrix} x_1 & x_2 & \cdots & x_n \end{bmatrix}^\top \in \mathbb{R}^{n \times d} \quad \text{and} \quad y = \begin{bmatrix} y_1 & y_2 & \cdots & y_n \end{bmatrix}^\top \in \mathbb{R}^n. \tag{10}$$

## 4.1 $L$-layer Linear Attention can Implement Degree-$O(3^L)$ Polynomials in $X^\top X$

We first present the following propositions to show that multi-layer as well as looped linear attention can be expressed as a polynomial function of $X^\top X$. This structure allows the models to leverage unlabeled data to improve the estimation of the task mean $\mu$.

**Proposition 1** *Given an L-layer linear attention model described in Section 2.2 with input prompt $Z$ defined in (3), one can construct the key, query, value weight matrices and the linear prediction head such that the model outputs (cf. (6))*

$$f_{att\text{-}L}(Z) = x^\top A X^\top y. \tag{11}$$

*Then, the following $A$ matrices are achievable via label and feature updates:*

- *Label propagation:* $A = c \prod_{\ell=1}^{L-1} (I + c_\ell X^\top X)$ *for arbitrary constants* $\{c, c_1, \cdots, c_{L-1}\}$;

- *Feature propagation:* $A = c (X^\top X)^{3^{L-1}-1}$ *for an arbitrary constant c.*

**Proposition 2** *Consider the same setting as in Proposition 1. There exists a single-layer linear attention model whose parameters can be constructed such that, when looped L times, its output reproduces that of (11), with $c_\ell \equiv c'$ for some arbitrary constant $c'$.*

The proofs of Proposition 1 and 2 are deferred to Appendix C.1 and C.2. In the following, we provide further clarification on the label and feature propagation.

1. The final prediction of the label propagation process can be rewritten as

$$f_{att\text{-}L}(Z) = c x^\top X^\top y_L \quad \text{where} \quad y_{\ell+1} = (I + c_\ell X X^\top) y_\ell, \quad \text{for} \quad \ell \in [L-1]$$

with $y_1 = y$. Here, $y_\ell$ can be interpreted as the soft pseudo-labels input to the $\ell$th layer, and each $c_\ell$ is parameterized by the attention mechanism in the corresponding layer. Although not

exactly equivalent, the *L*-layer linear attention process shares similarities with the Expectation-Maximization (EM) algorithm for semi-supervised learning, with *L* iterations of pseudo-labeling and a different label update strategy.

2. In contrast, the feature propagation process yields the final prediction

$$f_{\text{att-}L}(\boldsymbol{Z}) = c\boldsymbol{x}_L^\top \boldsymbol{X}_L^\top \boldsymbol{y} \quad \text{where} \quad \boldsymbol{X}_{\ell+1} = (\boldsymbol{X}_\ell \boldsymbol{X}_\ell^\top)\boldsymbol{X}_\ell \quad \text{and} \quad \boldsymbol{x}_{\ell+1} = (\boldsymbol{X}_\ell^\top \boldsymbol{X}_\ell)\boldsymbol{x}_\ell, \quad \text{for} \ \ell \in [L-1]$$

with $\boldsymbol{X}_1 = \boldsymbol{X}$ and $\boldsymbol{x}_1 = \boldsymbol{x}$. Here, $(\boldsymbol{X}_\ell, \boldsymbol{x}_\ell)$ can be viewed as the input features at the $\ell$th layer, encoding exponentially higher-order powers of $\boldsymbol{X}^\top \boldsymbol{X}$. This result highlights that a linear attention model requires only $O(\log K)$ layers to represent polynomial functions of degree $K$.

Our construction for *label propagation* is inherently related to the *gradient descent* emulation capability of linear attention Ahn et al. (2023). However, the *feature propagation* construction is fundamentally different and underscores the transformer's capability to implement rapid power iteration over the empirical covariance $\boldsymbol{X}^\top \boldsymbol{X}$. In the above constructions, each attention block with residual connections updates features or labels using one parameter, namely mappings of the form $\boldsymbol{X} \to \boldsymbol{X} + \alpha \boldsymbol{X}\boldsymbol{X}^\top \boldsymbol{X}$ or $\boldsymbol{y} \to \boldsymbol{y} + \beta \boldsymbol{X}\boldsymbol{X}^\top \boldsymbol{y}$. The lemma below shows that, even if the multilayer model can express polynomials of $\boldsymbol{X}^\top \boldsymbol{X}$ with exponential degrees in depth, the expressible manifold of polynomials has dimensionality linear in depth.

**Lemma 1 (Label + Feature Propagation)** *For an L-layer linear attention model, the resulting eventual prediction corresponds to the matrix $\boldsymbol{A}$ in Proposition 1 of the form*

$$\boldsymbol{A} = \sum_{\ell=0}^{(3^L-3)/2} a_\ell(\boldsymbol{X}^\top \boldsymbol{X})^\ell. \tag{12}$$

*The coefficients $\boldsymbol{a} := [a_0 \ a_1 \ \cdots \ a_{(3^L-3)/2}]^\top$ lie on a manifold of dimension at most $2L$ as $\boldsymbol{a}$ can be expressed as $\boldsymbol{a} = g(\boldsymbol{c})$ for some smooth function $g : \mathbb{R}^{2L} \to \mathbb{R}^{(3^L-3)/2}$ with $\boldsymbol{c}$ representing the parameters of individual layers.*

## 4.2 Which Semi-supervised Algorithm Does Multi-layer Attention Approximate?

Recall the SPI estimator $\hat{\boldsymbol{\mu}}_s$ from (SPI), and that $\boldsymbol{y}$ denotes the visible labels defined in Section 2.1 and (10). We have $\hat{\boldsymbol{\mu}}_s = \frac{1}{|\mathcal{I}|}\boldsymbol{X}^\top \boldsymbol{y}$. Motivated by Proposition 1 that multi-layer linear attention can implement higher-degree polynomials of $\boldsymbol{X}^\top \boldsymbol{X}$, we introduce the following SSPI estimator, which makes predictions based on the supervised estimate $\hat{\boldsymbol{\mu}}_s$ combined with higher-order debiased term of the form $(\boldsymbol{X}^\top \boldsymbol{X}/n - \sigma^2 \boldsymbol{I})^k$.

**Semisupervised Plug-in (SSPI) Estimator** Observe that the feature covariance satisfies $\mathbb{E}[\boldsymbol{X}^\top \boldsymbol{X}]/n = \boldsymbol{\mu}\boldsymbol{\mu}^\top + \sigma^2 \boldsymbol{I}$, and the top eigenvector of the centered covariance matrix $(\boldsymbol{X}^\top \boldsymbol{X}/n - \sigma^2 \boldsymbol{I})$ asymptotically aligns with either $\boldsymbol{\mu}$ or $-\boldsymbol{\mu}$. Therefore, with a substantial amount of unlabeled data, we propose the semisupervised plug-in (SSPI) estimator as follows:

$$\hat{\boldsymbol{\mu}}_{ss\text{-}k} = \alpha\hat{\boldsymbol{\mu}}_s + (1-\alpha)(\boldsymbol{X}^\top \boldsymbol{X}/n - \sigma^2 \boldsymbol{I})^k \hat{\boldsymbol{\mu}}_s \tag{SSPI-$k$}$$

where $\hat{\boldsymbol{\mu}}_s$ is the SPI estimator (cf. (SPI)), and $\alpha \in [0, 1]$ controls the trade-off between the fully-supervised and semi-supervised estimators. The optimal choice of $\alpha$ depends on the problem parameters $n, d$ and $p$. Note that as $k \to \infty$, the term $(\boldsymbol{X}^\top \boldsymbol{X}/n - \sigma^2 \boldsymbol{I})^k$ converges (up to scaling) to a rank-one projection onto the top eigenvector of the debiased covariance matrix, effectively serving as an estimator for $\boldsymbol{\mu}$ (up to sign).

In Figure 1, we present the prediction accuracies of 2-layer and 5-layer linear attention models, shown by green and orange markers, respectively. We also evaluate the SSPI algorithm with varying $k$ values, where the green solid curve corresponds to SSPI-1, and the green dotted represents SSPI-$\infty$, both using their respective optimal choices of $\alpha$. Details on selecting the optimal $\alpha$ values are provided in Section A.1 and illustrated in Figure 2. The results reveal a close alignment between multi-layer linear attention and SSPI estimators. Notably, the 2-layer model outperforms SSPI-1, due to its ability to implement higher-degree polynomials of $\boldsymbol{X}^\top \boldsymbol{X}$ (cf. Proposition 1 and Equation (12)). When the sample size is sufficiently large (e.g., $n > 50$ in Figure 1b), the top eigenvector provides a more accurate estimate of the task mean, enabling SSPI-$\infty$ to achieve higher accuracy. Furthermore, since

the 5-layer model is capable of representing higher-order functions than the 2-layer model, it can better estimate the top eigenvector, resulting in performance that closely matches that of SSPI-∞.

In the following, we analyze the optimal classifier of the form $sgn(x^\top A \hat{\mu}_s)$ for a GMM, and provide insights into its behavior in the asymptotic regime as $n \to \infty$.

**Theorem 2** *Consider a binary GMM defined in Section 2.1 and suppose that $(x_i, y_i)_{i=1}^{n+1}$ is generated using a fixed $\mu$ following (2). Given matrix $A \in \mathbb{R}^{d \times d}$, define prediction*

$$\hat{y}_A = sgn(x^\top A \hat{\mu}_s).$$

*where $\hat{\mu}_s$ is the SPI estimator defined in (SPI). Let $\mathcal{A}^\star := \min_{A \in \mathbb{R}^{d \times d}} \mathbb{P}(\hat{y}_A \neq y)$ be its optimal solution set. Then, $\mu\mu^\top \in \mathcal{A}^\star$. Additionally, it obeys*

$$\mathbb{P}(\hat{y}_{\mu\mu^\top} \neq y) = \underbrace{Q(1/\sigma)}_{Bayes\ error} + Q(\sqrt{np}/\sigma) - 2Q(1/\sigma)Q(\sqrt{np}/\sigma). \tag{13}$$

Note that, $\mathbb{P}(\hat{y}_{\mu\mu^\top} \neq y)$ depends on $np$ and $\sigma$ only, regardless of $\mu$ and $d$.

**Theorem 3** *Let the prompt $Z$ be generated as described in Section 2.2, and consider an L-layer linear attention model with $L \geq 2$ and $n = \infty$. Additionally, let $\hat{\mu}_s$ be the SPI estimator defined in (SPI). There exist model constructions such that for any $Z$ following (3), its prediction satisfies*

$$y_{att\text{-}L}(Z) = sgn(x^\top \mu\mu^\top \hat{\mu}_s).$$

The proof follows directly from Proposition 1 (label propagation), which shows that multi-layer linear attention can output $x^\top (X^\top X/n - \sigma^2 I) \hat{\mu}_s$. As $n \to \infty$, the empirical covariance converges to its expectation, i.e., $X^\top X/n - \sigma^2 I \to \mu\mu^\top$. The results in Figure 1c validate Theorem 3, showing that as $n$ becomes large enough (i.e., $n = 10000$), the predictions from both 2-layer and 5-layer linear attention models, as well as the SSPI-1 and SSPI-∞ estimators, closely align with the classification error characterized in Theorem 2, depicted by the black dotted line.

Theorem 3 establishes that, with infinitely many unlabeled samples, an $L$-layer linear attention model (for $L \geq 2$) can implement the predictor characterized in Theorem 2 using the optimal choice of $A$, thereby achieving the classification error specified in (13). In the following, we shift to the non-asymptotic setting where $n$ is finite and analyze the model's performance in this regime.

**Theorem 4** *Let the prompt $Z$ be generated as described in Section 2.2. Consider an L-layer linear attention model with $L \geq 2$ and denote its optimal prediction as $y^\star_{att\text{-}L}(Z)$. Additionally, let $\hat{\mu}_s$ be the SPI estimator defined in (SPI). Suppose that the number of labeled samples satisfies $np \geq 8d\sigma^2$ and $n > O(d)$ is sufficiently large. Then, there exists a universal constant $C > 0$ such that the classification error satisfies*

$$\mathbb{P}(y^\star_{att\text{-}L}(Z) \neq y) \leq Q\left(\frac{1 - C\sqrt{d/n}}{\sigma}\right) + e^{-d}.$$

The proof is deferred to Appendix C.5. Note that when $n \gg d$, the classification error approaches the Bayes error, i.e., $\mathbb{P}(y^\star_{att\text{-}L}(Z) \neq y) \approx Q(1/\sigma)$.

### 4.3 Multi-layer Attention as Expectation Maximization and Belief Propagation

In Section 4.1, we discussed how multi-layer linear attention can express polynomial functions of $X^\top X$. Here, we further explore the connection between multi-layer attention and the Expectation Maximization (EM) algorithm for semi-supervised learning. Beyond linear attention, we also highlight key differences between linear and softmax-based attention mechanisms, particularly in how they implement labeling strategies analogous to those in the EM algorithm.

Consider the following construction of the $\ell$-th layer attention weights:

$$W_q = W_k = \begin{bmatrix} I_d & 0 \\ 0 & 0 \end{bmatrix} \quad \text{and} \quad W_v = \begin{bmatrix} 0 & 0 \\ 0 & c_\ell \end{bmatrix}.$$

We examine both linear and softmax attention mechanisms. Let $\mathbb{S}(\cdot)$ denotes the softmax operation that applies on the rows of a matrix. With this, the data update defined in (5) becomes:

$$\text{Linear attention:} \quad \boldsymbol{y}_{\ell+1} = \boldsymbol{y}_\ell + c_\ell \boldsymbol{X}\boldsymbol{X}^\top \boldsymbol{y}_\ell$$

$$\text{Softmax attention:} \quad \boldsymbol{y}_{\ell+1} = \boldsymbol{y}_\ell + c_\ell \mathbb{S}(\boldsymbol{X}\boldsymbol{X}^\top)\boldsymbol{y}_\ell$$

In the case of linear attention, given the pseudo-labels $\boldsymbol{y}_\ell = [y_1^\ell, y_2^\ell, \ldots, y_n^\ell]$ at layer $\ell$, the model estimates the task mean using the SPI algorithm (cf. (SPI)) as $\hat{\boldsymbol{\mu}}_\ell = \boldsymbol{X}^\top \boldsymbol{y}_\ell$. The attention then updates each pseudo-label through the residual rule:

$$y_i^\ell \to y_i^\ell + c_\ell \boldsymbol{x}_i^\top \hat{\boldsymbol{\mu}}_\ell,$$

where $c_\ell$ is a layer-specific coefficient. Owing to the linearity of this mechanism, the resulting pseudo-labeling strategy aligns with a linear EM-style update.

In contrast, softmax attention computes pairwise similarities via the softmax of dot products. Define $s_{ij} = \frac{e^{\boldsymbol{x}_i^\top \boldsymbol{x}_j}}{\sum_{j \le n} e^{\boldsymbol{x}_i^\top \boldsymbol{x}_j}}$, then each pseudo-label is updated via:

$$y_i^\ell \to y_i^\ell + c_\ell \sum_{j \le n} s_{ij} y_j^\ell.$$

This update is a nonlinear, similarity-weighted pseudo-labeling strategy which can also be viewed as *belief propagation*. The nonlinear nature highlights a key distinction between softmax and linear attention in how they emulate EM–like updates ($s_{ij} = \boldsymbol{x}_i^\top \boldsymbol{x}_j$ for linear attention).

## 5 Experiments

In Sections 3 and 4, we introduced Figure 1 and demonstrated its consistency with our theoretical results. In this section, we describe the experimental setup and implementation details. Motivated by Proposition 2, which suggests that looping can help leverage unlabeled data, Section 5.1 introduces an algorithm based on the TabPFN, showing how it can enhance prediction performance by incorporating a small amount of unlabeled data and iterative pseudo-labeling through model looping. Additionally, we present further empirical findings to investigate additional questions of interest in Section A.1.

**Experimental Setup**  Following Section 2, set $d = 10$ and noise level $\sigma = 1$. All models are trained using Adam optimizer with a learning rate of $10^{-3}$ for 40,000 epochs, with a batch size of 512. We use logistic loss in our experiments. Since our study focuses on the optimization landscape and model expressivity, and experiments are implemented via gradient descent, we repeat 10 trainings from random initialization and results are presented as the maximal test accuracy among those 10 trails.

### 5.1 Tabular Experiments

To investigate how model looping (Proposition 2) can improve label prediction, we propose the LoopTabFM algorithm that addresses unlabeled data by iteratively assigning pseudo-labels. More details of the algorithm are deferred to Section A.2 and Algorithm 1.[1]

We evaluated the effectiveness of our proposed looping strategy by iteratively applying TabPFN-v2 on real-world binary classification benchmarks used in Hollmann et al. (2025). The results are summarized in Table 1, where each entry represents an average over 100 random splits of the dataset, with 80% of the data used as the test set in each split.

For each experiment, we randomly sample 10 labeled and 10 unlabeled examples, ensuring that the labeled set includes at least one example from each class. As a baseline (Loop-0), we apply TabPFN-v2 using only the labeled data. The corresponding test accuracies are reported in the "Loop-0" column of Table 1. We compare this to models updated through up to $k \le 5$ iterations of pseudo-label update, with results shown in the "Loop-$k$" columns. The final column reports the relative improvement (Rel. Imp.) over the baseline. Our results demonstrate that the looping strategy can significantly improve test accuracy. For instance, on OpenML datasets 1049, 1464, 40701, and 40983, accuracy improves by more than 10% over the baseline using only 10 additional unlabeled samples. The last row of

---

[1]Our code is available at `https://github.com/xiaofengliu-water/LoopTabFM`.

Table 1: Comparison of test accuracy (%) between the baseline (Loop-0) and LoopTabFM (Algorithm 1) after 1 to 5 iterations using TabPFN-v2. Each result is averaged over 100 random trials. The highest test accuracy for each dataset is highlighted in bold. The final column reports the relative improvement (%) of Loop-5 over the baseline, computed as (Loop-5 − Loop-0)/Loop-0×100%. Positive signs indicate a performance improvement over the baseline, while negative signs indicate a performance drop.

| OpenML ID | # of features | # of samples | Class imbalance | Loop-0 | Loop-1 | Loop-2 | Loop-3 | Loop-4 | Loop-5 | Rel. Imp. (%) |
|---|---|---|---|---|---|---|---|---|---|---|
| 3 | 36 | 3196 | 1.09 | 58.62 | 58.63 | 58.45 | 58.69 | **59.00** | 58.97 | 0.60 (+) |
| 31 | 20 | 1000 | 2.33 | **66.18** | 65.95 | 66.05 | 65.58 | 65.52 | 65.07 | 1.68 (−) |
| 1049 | 37 | 1458 | 7.19 | 72.00 | 75.62 | 79.48 | 80.31 | **81.49** | 81.40 | 13.06 (+) |
| 1067 | 21 | 2109 | 5.47 | 73.12 | 76.59 | 77.94 | 77.92 | 78.57 | **78.60** | 7.50 (+) |
| 1464 | 4 | 748 | 3.20 | 60.46 | 63.96 | 70.20 | 71.29 | **72.26** | 72.18 | 19.38 (+) |
| 1487 | 72 | 2534 | 14.84 | 82.54 | 87.67 | 88.57 | 88.27 | **89.85** | 89.56 | 8.51 (+) |
| 1489 | 5 | 5404 | 2.41 | 66.40 | 67.62 | **68.30** | 68.14 | 68.21 | 68.18 | 2.69 (+) |
| 1494 | 41 | 1055 | 1.96 | 62.24 | 63.05 | 64.62 | 65.94 | **66.07** | 66.05 | 6.12 (+) |
| 40701 | 20 | 5000 | 6.07 | 66.45 | 70.65 | 75.99 | **78.18** | 78.00 | 77.70 | 16.93 (+) |
| 40900 | 36 | 5100 | 67 | **98.53** | 98.41 | 98.39 | 98.39 | 98.27 | 98.26 | 0.28 (−) |
| 40981 | 14 | 690 | 1.25 | 73.56 | 74.41 | 74.67 | **74.99** | 74.93 | 74.94 | 1.88 (+) |
| 40983 | 5 | 4839 | 17.54 | 79.71 | 85.04 | 89.36 | **92.94** | 92.90 | 92.75 | 16.35 (+) |
| 41143 | 144 | 2984 | 1 | 64.64 | 64.80 | 65.06 | 65.17 | **65.29** | 65.13 | 0.76 (+) |
| 41144 | 259 | 3140 | 1.01 | 50.70 | 50.63 | 50.68 | 50.67 | 50.71 | **50.77** | 0.14 (+) |
| 41145 | 308 | 5832 | 1 | 56.16 | **56.28** | 56.21 | 56.24 | 56.19 | 56.22 | 0.12 (+) |
| 41146 | 20 | 5124 | 1 | 71.26 | 73.90 | 75.39 | 75.84 | 76.02 | **77.07** | 8.51 (+) |
| 41156 | 48 | 4147 | 3.03 | 67.74 | 69.78 | 70.64 | **71.82** | 71.72 | 71.74 | 5.90 (+) |
| **Average** | | | | 68.84 | 70.76 | 72.35 | 72.96 | **73.24** | 73.21 | 6.35 (+) |

the table reports average performance across datasets, revealing that the majority of performance gains occur in the first two iterations. This observation aligns with our synthetic experiments using multi-layer models (Figure 1), where the improvement from 1-layer to 2-layer is substantially greater than the improvement from 2-layer to 5-layer. These findings highlight that explicitly looping the tabular foundation model to iteratively refine soft pseudo-labels of unlabeled data using only a few iterations can substantially enhance performance.

As shown and discussed, our LoopTabFM algorithm enhances model performance. However, this improvement is not consistent across all datasets. For example, performance drops on the OpenML datasets with IDs 31 and 40900. This may be attributed to factors such as noise levels in the raw data, class imbalance, or other dataset-specific characteristics. In contrast to our synthetic experimental setting, where the model is pretrained in a meta-learning fashion on the distribution of the given dataset, TabPFN is used as a general-purpose pretrained foundation model and applied directly to target datasets in a single-shot inference setting. Prior work (Ye et al., 2025) has also shown that TabPFN can be sensitive to input length, which may further affect performance consistency. Despite these limitations, our experiments with TabPFN offer an initial insight into how unlabeled data and iterative looping can be leveraged to improve predictive performance. These findings suggest promising future directions, such as designing data-aware looping algorithms that adapt to dataset-specific properties.

# 6 Discussion and Limitations

Our paper introduces a theoretical study of semisupervised in-context learning and characterizes how transformer, specifically linear attention, models can harness unlabeled data in their context window to make inference. We show that depth is crucial to go beyond supervised estimation and utilize unlabeled data, and the latter is achieved by constructing estimators of the form $\hat{\mu} = \sum_{i=0}^{K} a_i (X^\top X)^i X^\top y$. log $K$ depth suffices to express a $K$th order polynomial which is in line with our synthetic and real experiments that corroborate that mild amount of depth/looping already achieves most of the benefit. Our core theoretical results are limited to linear attention models and it is important to understand the capabilities of the full transformer architecture. Indeed, transformer (MLP+softmax) empirically outperforms a linear attention model with equal number of layers, well approximating the Bayes optimal semisupervised estimator. It would also be exciting to go beyond the classification setting and examine how self-generated CoT rationales, as in (Wu et al., 2023), can enhance ICL capabilities for tasks that require reasoning/autoregression. Additionally, our proposed LoopTabFM algorithm demonstrates that iteratively pseudo-labeling unlabeled data can indeed enhance predictive performance for tabular tasks. However, there remains significant potential for developing more intelligent, data-specific algorithms that more effectively leverage unlabeled data to further improve model performance.

## Acknowledgements

This work was supported in part by the National Science Foundation grants CCF-2046816, CCF-2403075, CCF-2008020, the Office of Naval Research grant N000142412289, and by gifts/awards from Open Philanthropy, Amazon Research, and Google Research.

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

# Appendix

## Table of Contents

(a) $n = 50$, $p \in (0, 1]$.    (b) $np = 10$, $n \in \{10, \cdots, 10000\}$.    (c) $n = 50$, $p \in (0, 1]$.

Figure 2: Additional experimental results. (a)&(b): Analysis of the optimal $\alpha$ values for the SSPI estimator (cf. (SSPI-$k$)) under varying $(n, p, k)$. Green solid and dotted curves represent optimal $\alpha$ values for SSPI-1 and SSPI-$\infty$, respectively. The SSPI results shown in Figure 1 use the corresponding $\alpha$ values from Figs. 2a and 2b. (c): Comparison of different model architectures for the SS-ICL problem. Dark blue and orange curves show results for 1-layer and 5-layer attention models, with solid and dashed curves representing linear and softmax attention, respectively. Cyan curves correspond to 5-layer Transformers. The black dotted curve shows the asymptotic Bayes-optimal error (cf. Lelarge & Miolane (2019)). Results suggest the performance ordering: Transformer > linear attention > softmax attention. Further details are provided in Section 5.

## A  Additional Experiments and Algorithmic Details

### A.1  Additional Observations

**Exploration of Optimal $\alpha$ Values**    In Section 4, we introduced the SSPI-$k$ estimator (cf. (SSPI-$k$)), but did not discuss the choice of the mixing parameter $\alpha$, which plays a crucial role in balancing the contribution of the supervised estimator $\hat{\mu}_s$. Specifically, $\alpha$ controls how much weight is given to the purely supervised signal. In the fully supervised case, the optimal choice is $\alpha = 1$, as $\hat{\mu}_s$ corresponds to the optimal estimator.

In Figures 2a and 2b, we empirically examine the optimal values of $\alpha$. Given $\boldsymbol{\mu} \sim \mathrm{Unif}(\mathbb{S}^{d-1})$, we define the optimal $\alpha$ as the minimizer of the following cosine similarity-based objective:

$$\alpha^\star := \min_{\alpha \in [0,1]} \mathcal{L}(\alpha) \quad \text{where} \quad \mathcal{L}(\alpha) = 1 - \mathbb{E}[\texttt{cosine\_similarity}(\boldsymbol{\mu}_{ss\text{-}k}, \boldsymbol{\mu})].$$

**Algorithm 1 LoopTabFM**: Looping Tabular FM with Soft Pseudo-labels and Risk-aware Updates

**Require:** Dataset $\mathcal{D}_{\text{lab}}, \mathcal{D}_{\text{unlab}}$, looping iterations $K$
1: **procedure** LOOPING($\mathcal{D}_{\text{lab}}, \mathcal{D}_{\text{unlab}}, K$)
2:     $\text{FM}_0 \leftarrow \text{TabPFN-v2}(\mathcal{D}_{\text{lab}})$                    ▷ $\text{FM}_k$ corresponds to model of Loop-$k$.
3:     $\mathcal{D}_{\text{unlab}} \leftarrow \text{FM}_0(\mathcal{D}_{\text{unlab}})$       ▷ Assign pseudo labels via $\hat{y}^{\text{soft}} \leftarrow \text{FM}_0(\boldsymbol{x} \in \mathcal{D}_{\text{unlab}})$.
4:     $\text{FM}_{\text{best}} \leftarrow \text{FM}_0$
5:     $\mathcal{R}_{\text{val}} = \text{Val\_Risk}(\mathcal{D}_{\text{unlab}})$
6:     **for** Looping iteration $k = 1, \ldots, K$ **do**
7:         $\text{FM}_k \leftarrow \text{TabPFN-v2}(\mathcal{D}_{\text{lab}} \cup \mathcal{D}_{\text{unlab}})$
8:         $\mathcal{D}_{\text{unlab}} \leftarrow \text{FM}_k(\mathcal{D}_{\text{unlab}})$    ▷ Update pseudo labels via $\hat{y}^{\text{soft}} \leftarrow \text{FM}_k(\boldsymbol{x} \in \mathcal{D}_{\text{unlab}})$.
9:         **if** $\text{Val\_Risk}(\mathcal{D}_{\text{unlab}}) < \mathcal{R}_{\text{val}}$ **then**
10:             $\text{FM}_{\text{best}} \leftarrow \text{FM}_k$
11:             $\mathcal{R}_{\text{val}} = \text{Val\_Risk}(\mathcal{D}_{\text{unlab}})$
12:        **end if**
13:    **end for**
14:    **return** $\text{FM}_{\text{best}}$
15: **end procedure**
16: **procedure** VAL\_RISK($\mathcal{D}_{\text{unlab}}$)
17:    **return** $\frac{1}{|\mathcal{D}_{\text{unlab}}|} \sum_i \min(|\hat{y}_i^{\text{soft}} - 1|, |\hat{y}_i^{\text{soft}} + 1|)$
18:                    ▷ $\hat{y}^{\text{soft}}$ corresponds to the assigned soft label for feature in $\mathcal{D}_{\text{unlab}}$.
19: **end procedure**

For each setting, we optimize $\alpha$ using the Adam optimizer for 10,000 epochs with a batch size of 128 and a learning rate of 0.01. The results are shown in Figs 2a and 2b.

In Figure 2a, for both SSPI-1 and SSPI-$\infty$, the optimal $\alpha$ starts near zero when the number of labeled examples is small, reflecting the limited utility of $\hat{\boldsymbol{\mu}}_s$ in low-supervision regimes. As the number of labeled samples increases, $\alpha$ grows approximately linearly and approaches 1 when the problem becomes fully supervised. In Figure 2b, when $n = 10$ and $p = 1$ (i.e., all examples are labeled), the optimal $\alpha$ begins at 1. As $n$ increases and the fraction of unlabeled data grows, $\alpha$ decreases significantly. This trend indicates that as the volume of unlabeled data increases, the SSPI estimator adaptively reduces reliance on the supervised component $\hat{\boldsymbol{\mu}}_s$ and increases reliance on the semi-supervised component, which leverages the structure of the unlabeled data through $\boldsymbol{X}^\top \boldsymbol{X}$.

**Comparison Across Different Model Architectures** Beyond linear attention, we investigate additional model architectures under our SS-ICL setting. The comparison results are presented in Fig. 2c. The softmax attention model uses the same structure described in Section 2.2, with the only difference being the addition of a softmax operation in Eq. (4). The Transformer model introduces further nonlinearity and capacity by incorporating multi-layer perceptrons (MLPs) and layer normalization. The Transformer experiments are conducted with 5-layer models.

When comparing weaker models—such as 1-layer linear (dark blue solid) and softmax (dark blue dashed) attention—we observe that softmax attention consistently underperforms linear attention. Notably, softmax attention fails to match the performance of the optimal supervised estimator, even when all labels are observed (i.e., when the number of labeled samples equals $n = 50$). Furthermore, increasing the depth of softmax attention (orange dashed curve for 5-layer softmax) still does not surpass the performance of 5-layer linear attention (orange solid curve). Among all architectures, the Transformer achieves the best performance due to its increased model capacity and expressiveness. Compared with Fig. 1a, where the orange and dark blue markers (linear attention) are identical, the Transformer significantly improves accuracy. This improvement highlights that SSPI, while effective, is not the optimal semi-supervised estimator. Although our semi-supervised setting assumes isotropic data, the characterization of its optimal algorithm remains an open and foundational problem for future exploration. In the figure, we also include the asymptotic Bayes-optimal curve (black dotted; derived from Lelarge & Miolane (2019)) . As the number of samples increases, the results from linear attention, softmax attention, and Transformer all converge toward this optimal curve. We attribute the initial performance gap, particularly at low values along $x$-axis (e.g., $np = 1$), to the scarcity of labeled data.

## A.2 Algorithmic Details of Tabular Experiments

In this section, we provide additional details regarding the tabular experiments discussed in Section 5.1. We propose the LoopTabFM algorithm with its details outlined in Algorithm 1. Suppose that we are given labeled $\mathcal{D}_{\text{lab}}$ and unlabeled $\mathcal{D}_{\text{unlab}}$ datasets. The overall workflow of the algorithm proceeds as follows:

1. **Base Model:** Perform ICL using TabPFN on the labeled dataset $\mathcal{D}_{\text{lab}}$ and treat the resulting model as the base model (Loop-0). The corresponding test accuracies are reported in Table 1.

2. **Pseudo-Label Assignment:** Using the current model (e.g., Loop-$k$) to generate predictions for the unlabeled data $\mathcal{D}_{\text{unlab}}$. Assign soft pseudo-labels based on these predictions. Note that the model outputs are scalars (i.e., elements of $\mathbb{R}$) and can be interpreted as soft labels.

3. **Model Update:** Construct a new prompt by combining the labeled examples with their true labels and the unlabeled examples with their assigned soft pseudo-labels. Perform ICL using TabPFN on this combined prompt to obtain an updated model (Loop-$(k + 1)$). Repeat this process from Step 2 until the maximum number of looping iterations is reached.

⋆ **Model Validation:** To improve the stability of the looping process, we introduce an additional validation step and retain the model with the lowest validation risk as the final (best) model. Specifically, after assigning soft pseudo-labels to the unlabeled data, i.e., $\mathcal{D}_{\text{unlab}} = \{(x_i, \hat{y}_i^{\text{soft}})_{i=1}^n\}$, we compute the validation risk over these pseudo-labeled examples as follows:

$$\texttt{Val\_Risk}(\mathcal{D}_{\text{unlab}}) = \frac{1}{n} \sum_{i \in [n]} \min\left(\left|\hat{y}_i^{\text{soft}} - 1\right|, \left|\hat{y}_i^{\text{soft}} + 1\right|\right),$$

which penalizes predictions that deviate from confident binary labels $\pm 1$.

# B  Analysis of Single-layer Linear Attention

## B.1  Supporting Lemmas

Recap the SPI estimator from (SPI). Given a semi-supervised dataset $(x_i, y_i)_{i=1}^n$ as described in Section 2.1, let $\mathcal{I}$ denote the token indices set corresponding to the labeled demonstrations, that is, we have

$$y_i = \begin{cases} y_i^c, & i \in \mathcal{I} \\ 0, & otherwise. \end{cases} \tag{14}$$

Then, the SPI estimates the task mean via

$$\hat{\boldsymbol{\mu}}_s = \frac{1}{|\mathcal{I}|} \sum_{i \in \mathcal{I}} y_i x_i.$$

Let $W \in \mathbb{R}^{d \times d}$ be the preconditioning matrix. We define the following objective:

$$W^\star := \arg\min_{W \in \mathbb{R}^{d \times d}} \tilde{\mathcal{L}}(W) \quad \text{where} \quad \tilde{\mathcal{L}}(W) = \mathbb{E}\left[\left(x^\top W \sum_{i \in \mathcal{I}} y_i x_i - y\right)^2\right]. \tag{15}$$

Here, we set $(x, y)$ to be the query feature and its corresponding true label. The expectation subsumes the randomness in $(x_i, y_i), (x, y)$ as described in Section 2.1.

In the following, we provide a lemma that establishes equivalence between optimizing $\mathcal{L}_{\text{att-1}}(W^{(1)}, h)$ (cf. (7) and choosing $L = 1$) and $\tilde{\mathcal{L}}(W)$.

**Lemma 2** *Consider ICL problem described in Section 2.2 with prompt defined in (3). Consider training a single-layer linear attention with squared loss, that is, $L = 1$ and $\ell(y, \hat{y}) = (y - \hat{y})^2$. Recall the objectives from (7) and (15), and let $\mathcal{L}_{\text{att-1}}^\star$ and $\tilde{\mathcal{L}}^\star := \tilde{\mathcal{L}}(W^\star)$ be their corresponding optimal losses where $W^\star$ is defined in (15). Then, we have*

$$\mathcal{L}_{att\text{-}1}^\star = \tilde{\mathcal{L}}^\star. \tag{16}$$

Additionally, let $f^\star_{\text{att-1}} : \mathbb{R}^{(n+1)\times(d+1)} \to \mathbb{R}$ denote the optimal prediction (associated with the optimal loss $\mathcal{L}^\star_{\text{att-1}}$). We have that $f^\star_{\text{att-1}}$ is unique and for any prompt $\mathbf{Z}$ (cf. (3))

$$f^\star_{\text{att-1}}(\mathbf{Z}) = \mathbf{x}^\top \mathbf{W}^\star \sum_{i \in \mathcal{I}} y_i \mathbf{x}_i. \tag{17}$$

**Proof.** Recap the single-layer linear attention model and its prediction from (4) and (6). We have

$$f_{\text{att-1}}(\mathbf{Z}) = \mathbf{h}^\top \text{att}(\mathbf{Z}; \mathcal{W})_{[n+1]} \quad \text{where} \quad \text{att}(\mathbf{Z}; \mathcal{W}) = (\mathbf{Z} \mathbf{W}_q \mathbf{W}_k^\top \mathbf{Z}^\top) \mathbf{M} \mathbf{Z} \mathbf{W}_v \tag{18}$$

with $\mathcal{W} := \{\mathbf{W}_q, \mathbf{W}_k, \mathbf{W}_v\}$ being the set of the query, key and value matrices of the attention. Since $\mathcal{W}$ and $\mathbf{h}$ are tunable parameters, without loss of generality and for simplicity, let

$$\mathbf{W} := \mathbf{W}_q \mathbf{W}_k^\top \quad \text{and} \quad \bar{\mathbf{h}} := \mathbf{W}_v \mathbf{h}.$$

Following the proof of Li et al., 2024, Proposition 1, similarly, we denote

$$\mathbf{W} = \begin{bmatrix} \bar{\mathbf{W}} & \mathbf{w}_1 \\ \mathbf{w}_2^\top & w \end{bmatrix} \quad \text{and} \quad \bar{\mathbf{h}} = \begin{bmatrix} \mathbf{h}_1 \\ h \end{bmatrix},$$

where $\bar{\mathbf{W}} \in \mathbb{R}^{d\times d}$, $\mathbf{w}_1, \mathbf{w}_2, \mathbf{h}_1 \in \mathbb{R}^d$, and $w, h \in \mathbb{R}$.

Additionally, let $\mathcal{I}$ denote the token indices set corresponding to the labeled demonstrations (cf. (14)). Recall the prompt $\mathbf{Z}$ from (3), and $\mathbf{X} = [\mathbf{x}_1 \cdots \mathbf{x}_n]^\top \in \mathbb{R}^{n\times d}$ and $\mathbf{y} = [y_1 \cdots y_n]^\top \in \mathbb{R}^n$ from (10). Then we get

$$\mathbf{Z} = \begin{bmatrix} \mathbf{x}_1 & \mathbf{x}_2 & \cdots & \mathbf{x}_n & \mathbf{x} \\ y_1 & y_2 & \cdots & y_n & 0 \end{bmatrix}^\top = \begin{bmatrix} \mathbf{X}^\top & \mathbf{x} \\ \mathbf{y}^\top & 0 \end{bmatrix}^\top \in \mathbb{R}^{(n+1)\times(d+1)}. \tag{19}$$

Combining (18) and (19) together, we can rewrite the one-layer linear prediction as

$$\begin{aligned}
f_{\text{att-1}}(\mathbf{Z}) &= [\mathbf{x}^\top \ 0] \mathbf{W} \mathbf{Z}^\top \mathbf{M} \mathbf{Z} \bar{\mathbf{h}} \\
&= [\mathbf{x}^\top \ 0] \begin{bmatrix} \bar{\mathbf{W}} & \mathbf{w}_1 \\ \mathbf{w}_2^\top & w \end{bmatrix} \begin{bmatrix} \mathbf{X}^\top & \mathbf{x} \\ \mathbf{y}^\top & 0 \end{bmatrix} \begin{bmatrix} \mathbf{I}_n & 0 \\ 0 & 0 \end{bmatrix} \begin{bmatrix} \mathbf{X}^\top & \mathbf{x} \\ \mathbf{y}^\top & 0 \end{bmatrix}^\top \begin{bmatrix} \mathbf{h}_1 \\ h \end{bmatrix} \\
&= [\mathbf{x}^\top \bar{\mathbf{W}} \ \ \mathbf{x}^\top \mathbf{w}_1] \begin{bmatrix} \mathbf{X}^\top \mathbf{X} & \mathbf{X}^\top \mathbf{y} \\ \mathbf{y}^\top \mathbf{X} & \mathbf{y}^\top \mathbf{y} \end{bmatrix} \begin{bmatrix} \mathbf{h}_1 \\ h \end{bmatrix} \\
&= [\mathbf{x}^\top \bar{\mathbf{W}} \ \ \mathbf{x}^\top \mathbf{w}_1] \begin{bmatrix} \mathbf{X}^\top \mathbf{X} \mathbf{h}_1 + h \mathbf{X}^\top \mathbf{y} \\ \mathbf{y}^\top \mathbf{X} \mathbf{h}_1 + h \mathbf{y}^\top \mathbf{y} \end{bmatrix} \\
&= \mathbf{x}^\top \bar{\mathbf{W}} (\mathbf{X}^\top \mathbf{X} \mathbf{h}_1 + h \mathbf{X}^\top \mathbf{y}) + \mathbf{x}^\top \mathbf{w}_1 (\mathbf{y}^\top \mathbf{X} \mathbf{h}_1 + h \mathbf{y}^\top \mathbf{y}) \\
&= \mathbf{x}^\top (h \bar{\mathbf{W}} + \mathbf{w}_1 \mathbf{h}_1^\top) \mathbf{X}^\top \mathbf{y} + \mathbf{x}^\top (\bar{\mathbf{W}} \mathbf{X}^\top \mathbf{X} \mathbf{h}_1 + h \mathbf{y}^\top \mathbf{y} \mathbf{w}_1) \\
&= \mathbf{x}^\top \tilde{\mathbf{W}} \mathbf{X}^\top \mathbf{y} + \mathbf{x}^\top (\bar{\mathbf{W}} \mathbf{X}^\top \mathbf{X} \mathbf{h}_1 + mhw_1)
\end{aligned}$$

where $\tilde{\mathbf{W}} := h \bar{\mathbf{W}} + \mathbf{w}_1 \mathbf{h}_1^\top$ and we define $m := |\mathcal{I}|$.

Next, recall the loss from (7) and consider the squared loss function, $\ell(y, \hat{y}) = (y - \hat{y})^2$. We have

$$\begin{aligned}
\mathcal{L}_{\text{att-1}}(\mathcal{W}^{(1)}, \mathbf{h}) &= \mathbb{E}\left[(f_{\text{att-1}}(\mathbf{Z}) - y)^2\right] \\
&= \mathbb{E}\left[\left(\mathbf{x}^\top \tilde{\mathbf{W}} \mathbf{X} \mathbf{y} + \mathbf{x}^\top \left(\bar{\mathbf{W}} \mathbf{X}^\top \mathbf{X} \mathbf{h}_1 + mhw_1\right) - y\right)^2\right] \\
&= \mathbb{E}\left[\left(y\mathbf{x}^\top \tilde{\mathbf{W}} \mathbf{X} \mathbf{y} + y\mathbf{x}^\top \left(\bar{\mathbf{W}} \mathbf{X}^\top \mathbf{X} \mathbf{h}_1 + mhw_1\right) - 1\right)^2\right].
\end{aligned}$$

For simplicity and without loss of generality, we omit $y$ and use $\mathbf{x}$ to represent $y\mathbf{x}$. Note that the distribution of (updated) $\mathbf{x}$ is not conditioned on its class and given mean vector $\boldsymbol{\mu}$, it follows $\mathbf{x} \sim \mathcal{N}(\boldsymbol{\mu}, \sigma^2 \mathbf{I})$. Similarly, let $\mathbf{x}_i$ represent $y_i^c \mathbf{x}_i$. We can then write

$$\begin{aligned}
\mathcal{L}_{\text{att-1}}(\mathcal{W}^{(1)}, \mathbf{h}) &= \mathbb{E}\left[\left(\mathbf{x}^\top \tilde{\mathbf{W}} \sum_{i\in\mathcal{I}} \mathbf{x}_i + \mathbf{x}^\top \left(\bar{\mathbf{W}} \mathbf{X}^\top \mathbf{X} \mathbf{h}_1 + mhw_1\right) - 1\right)^2\right] \tag{20} \\
&= \mathbb{E}\left[\left(\mathbf{x}^\top \tilde{\mathbf{W}} \sum_{i\in\mathcal{I}} \mathbf{x}_i - 1\right)^2\right] + \mathbb{E}\left[\left(\mathbf{x}^\top \left(\bar{\mathbf{W}} \mathbf{X}^\top \mathbf{X} \mathbf{h}_1 + mhw_1\right)\right)^2\right] \\
&\quad + 2\mathbb{E}\left[\left(\mathbf{x}^\top \tilde{\mathbf{W}} \sum_{i\in\mathcal{I}} \mathbf{x}_i - 1\right)\left(\mathbf{x}^\top \left(\bar{\mathbf{W}} \mathbf{X}^\top \mathbf{X} \mathbf{h}_1 + mhw_1\right)\right)\right].
\end{aligned}$$

We start with showing that for any given parameters $W \in \mathbb{R}^{(d+1)\times(d+1)}, h \in \mathbb{R}^{d+1}$, the component $\mathbb{E}[(x^\top \tilde{W} \sum_{i\in I} x_i - 1)(x^\top (\bar{W} X^\top X h_1 + mh w_1))] = 0$. To prove it, we first expand

$$(x^\top \tilde{W} \sum_{i\in I} x_i - 1)(x^\top (\bar{W} X^\top X h_1 + mh w_1))$$

$$= \underbrace{(x^\top \tilde{W} \sum_{i\in I} x_i)(x^\top \bar{W} X^\top X h_1)}_{(a)} - \underbrace{x^\top \bar{W} X^\top X h_1}_{(b)} + \underbrace{(x^\top \tilde{W} \sum_{i\in I} x_i)(mh x^\top w_1)}_{(c)} - \underbrace{mh x^\top w_1}_{(d)}.$$

In the following, we consider the expectations of $(a), (b), (c), (d)$ sequentially, all of which take the value zero. First note that since $\mu \sim \text{Unif}(\mathbb{S}^{d-1})$ and $(\xi_i)_{i=1}^n, \xi \sim \mathcal{N}(0, \sigma^2 I)$, the odd moments of $\mu, \xi$ and $\xi_i, i \in [n]$ are all zeros.

$(a):$
$$\mathbb{E}\left[(x^\top \tilde{W} \sum_{i\in I} x_i)(x^\top \bar{W} X^\top X h_1)\right]$$

$$= \mathbb{E}\left[(\mu + \xi)^\top \tilde{W} \sum_{i\in I}(\mu + \xi_i)(\mu + \xi)^\top \bar{W} \sum_{i\in[n]}(\mu + \xi_i)(\mu + \xi_i)^\top h_1\right]$$

$$= \sum_{i\in I}\sum_{j\in[n]} \mathbb{E}\left[(\mu + \xi)^\top \tilde{W}(\mu + \xi_i)(\mu + \xi)^\top \bar{W}(\mu + \xi_j)(\mu + \xi_j)^\top h_1\right]$$

$$= \sum_{i\in I}\sum_{j\in[n]} \mathbb{E}\left[\mu^\top \tilde{W}\mu\mu^\top \bar{W}(\mu\mu^\top + \xi_j \xi_j^\top)h_1 + \xi^\top \tilde{W}\mu\xi^\top \bar{W}(\mu\mu^\top + \xi_j \xi_j^\top)h_1\right]$$

$$= 0,$$

$(b):$
$$\mathbb{E}\left[x^\top \bar{W} X^\top X h_1\right]$$

$$= \mathbb{E}\left[(\mu + \xi)^\top \bar{W} \sum_{i\in[n]}(\mu + \xi_i)(\mu + \xi_i)^\top h_1\right]$$

$$= \mathbb{E}\left[\mu^\top \bar{W} \sum_{i\in[n]}(\mu\mu^\top + \xi_i \xi_i^\top)h_1\right]$$

$$= 0,$$

$(c):$
$$\mathbb{E}\left[(x^\top \tilde{W} \sum_{i\in I} x_i)(mh x^\top w_1)\right]$$

$$= mh\, \mathbb{E}\left[(\mu + \xi)^\top \tilde{W} \sum_{i\in I}(\mu + \xi_i)(\mu + \xi)^\top w_1\right]$$

$$= mh \sum_{i\in I} \mathbb{E}\left[(\mu + \xi)^\top \tilde{W}\mu(\mu + \xi)^\top w_1\right]$$

$$= mh \sum_{i\in I} \mathbb{E}\left[\mu^\top \tilde{W}\mu\mu^\top w_1 + \xi^\top \tilde{W}\mu\xi^\top w_1\right]$$

$$= 0,$$

$(d):$ $\mathbb{E}\left[mh x^\top w_1\right] = 0.$

Therefore, loss in (20) returns

$$\mathcal{L}_{\text{att-1}}(W^{(1)}, h) = \underbrace{\mathbb{E}\left[\left(x^\top \tilde{W} \sum_{i\in I} x_i - 1\right)^2\right]}_{\tilde{\mathcal{L}}(\bar{W})} + \mathbb{E}\left[\left(x^\top (\bar{W} X^\top X h_1 + mh w_1)\right)^2\right].$$

Here, the first term $\mathbb{E}[(x^\top \tilde{W} \sum_{i \in I} x_i - 1)^2] = \tilde{\mathcal{L}}(\tilde{W})$ where $\tilde{\mathcal{L}}(\tilde{W})$ is defined in (15).

Recall that $\tilde{W} = h\bar{W} + w_1 h_1^\top$. Then for any $\tilde{W} \in \mathbb{R}^{d \times d}$, setting $h_1 = w_1 = \mathbf{0}_d$ and $h = 1$ returns $\mathbb{E}\left[\left(x^\top \left(\bar{W} X^\top X h_1 + mhw_1\right)\right)^2\right] = 0$, and then

$$\mathcal{L}_{\text{att-1}}(\mathcal{W}^{(1)}, h) = \mathbb{E}\left[\left(x^\top \bar{W} \sum_{i \in I} x_i - 1\right)^2\right]$$

Therefore, optimizing $\mathcal{L}_{\text{att-1}}(\mathcal{W}^{(1)}, h)$ returns the same minima as optimizing $\tilde{\mathcal{L}}(W)$, which completes the proof of (16). Note that optimal loss $\mathcal{L}^\star_{\text{att-1}}$ depends on the labeled data $i \in I$ only.

Furthermore, since $\tilde{\mathcal{L}}(W)$ is strongly convex (see (21)), $W^\star$ exists and is unique. Therefore, (16) and uniqueness of $W^\star$ leads to the conclusion (17). ∎

**Lemma 3** *Consider the objective defined in* (15) *with semi-supervised data following Section 2. Then the optimal solution* $W^\star$ *satisfies*
$$W^\star = cI$$
*for some* $c > 0$.

**Proof.** Recap the Objective (15) and its optimal solution $W^\star$. Let $I$ be the index set corresponding the labeled in-context examples, and $|I| = m$. Note that, $m$ is also a random variable, independent of $x_i, y_i^c, x, y$.

As in the proof of Lemma 2, we use $x$ to represent $yx$ and $x_i$ to represent $y_i^c x_i$ for simplicity, where (updated) $x_i, x \sim \mathcal{N}(\mu, \sigma^2 I)$. Letting $\xi', \xi, \xi_i \sim \mathcal{N}(0, \sigma^2 I)$ be independent, we obtain

$$\tilde{\mathcal{L}}(W) = \mathbb{E}\left[(x^\top W \sum_{i \in I} x_i - 1)^2\right] \tag{21}$$

$$= \mathbb{E}\left[((\mu + \xi)^\top W \sum_{i \in I}(\mu + \xi_i) - 1)^2\right]$$

$$= \mathbb{E}\left[((\mu + \xi)^\top W(m\mu + \sqrt{m}\xi') - 1)^2\right]$$

$$= \mathbb{E}\left[m^2(\mu^\top W\mu)^2 + m(\mu^\top W\xi')^2 + m^2(\xi^\top W\mu)^2 + m(\xi^\top W\xi')^2 + 1\right] - 2\mathbb{E}\left[m\mu^\top W\mu\right]$$

$$= \frac{\mathbb{E}[m^2]}{d(d+2)}(\text{tr}(W)^2 + \text{tr}(WW^\top) + \text{tr}(W^2)) + \frac{\mathbb{E}[m+m^2]}{d}\sigma^2\text{tr}(WW^\top)$$

$$+ \mathbb{E}[m]\sigma^4\text{tr}(WW^\top) + 1 - \frac{2\mathbb{E}[m]}{d}\text{tr}(W).$$

Differentiating it results in

$$\nabla_W \tilde{\mathcal{L}}(W) = \frac{2\mathbb{E}[m^2]}{d(d+2)}(\text{tr}(W)I + W + W^\top) + \frac{2\mathbb{E}[m+m^2]\sigma^2}{d}W + 2\mathbb{E}[m]\sigma^4 W - \frac{2\mathbb{E}[m]}{d}I.$$

Setting $\nabla_W \tilde{\mathcal{L}}(W) = 0$, we obtain the optimal $W^\star$

$$W^\star = \frac{1}{(1+\sigma^2)\mathbb{E}[m^2]/\mathbb{E}[m] + \sigma^2 + \sigma^4 d}I,$$

which leads to the conclusion that $W^\star = cI$, for $c = \frac{1}{(1+\sigma^2)\mathbb{E}[m^2]/\mathbb{E}[m]+\sigma^2+\sigma^4 d} > 0$. It completes the proof. ∎

## B.2 Proof of Theorem 1

**Proof.** Note that (8) can be easily proven using Lemmas 2 and 3. Then, we focus on proving (9).

Given that (8) holds, we can rewrite its classification error as

$$\mathbb{P}(y^\star_{\text{att-1}}(Z) \neq y) = \mathbb{P}(\text{sgn}(x^\top \hat{\mu}_s) \neq y) = \mathbb{P}(\text{sgn}(yx^\top \hat{\mu}_s) \neq 1) \tag{22}$$

where $\hat{\boldsymbol{\mu}}_s = \frac{1}{|I|} \sum_{i \in I} y_i \boldsymbol{x}_i$ defined in (SPI) and $I$ is the index set of labeled samples. Let $m = |I|$.

Recall from Section 2.1 where $\boldsymbol{x} \sim \mathcal{N}(y \cdot \boldsymbol{\mu}, \sigma^2 \boldsymbol{I})$. We can rewrite

$$y\boldsymbol{x} = \boldsymbol{\mu} + \sigma \boldsymbol{g}_1 \quad \text{where} \quad \boldsymbol{g}_1 \sim \mathcal{N}(0, \boldsymbol{I}).$$

Then for any given $\boldsymbol{\mu}, \hat{\boldsymbol{\mu}}_s$, we get

$$
\begin{aligned}
\mathbb{P}\left(\mathrm{sgn}(y\boldsymbol{x}^\top \hat{\boldsymbol{\mu}}_s) \neq 1 \mid \boldsymbol{\mu}, \hat{\boldsymbol{\mu}}_s\right) &= \mathbb{P}\left((\boldsymbol{\mu} + \sigma \boldsymbol{g}_1)^\top \hat{\boldsymbol{\mu}}_s < 0 \mid \boldsymbol{\mu}, \hat{\boldsymbol{\mu}}_s\right) \\
&= \mathbb{P}\left(\boldsymbol{\mu}^\top \hat{\boldsymbol{\mu}}_s < \sigma \boldsymbol{g}_1^\top \hat{\boldsymbol{\mu}}_s \mid \boldsymbol{\mu}, \hat{\boldsymbol{\mu}}_s\right) \\
&= Q\left(\frac{\boldsymbol{\mu}^\top \hat{\boldsymbol{\mu}}_s}{\sigma \|\hat{\boldsymbol{\mu}}_s\|_{\ell_2}}\right).
\end{aligned}
\tag{23}
$$

Here $Q$-function is the tail distribution function of the standard normal distribution.

Next, similarly, given that $\boldsymbol{x}_i \sim \mathcal{N}(y_i \cdot \boldsymbol{\mu}, \sigma^2 \boldsymbol{I})$ for $i \in I$, we can rewrite

$$\hat{\boldsymbol{\mu}}_s = \frac{1}{m} \sum_{i \in I} y_i \boldsymbol{x}_i = \boldsymbol{\mu} + \frac{\sigma}{\sqrt{m}} \boldsymbol{g}_2 \quad \text{where} \quad \boldsymbol{g}_2 \sim \mathcal{N}(0, \boldsymbol{I}).$$

Then combining (22) and (23), we have

$$
\begin{aligned}
\mathbb{P}(y^\star_{\text{att-1}}(\boldsymbol{Z}) \neq y) &= \mathbb{E}_{\boldsymbol{\mu}, \boldsymbol{g}_2} \left[ Q\left(\frac{\boldsymbol{\mu}^\top \hat{\boldsymbol{\mu}}_s}{\sigma \|\hat{\boldsymbol{\mu}}_s\|_{\ell_2}}\right) \right] \\
&= \mathbb{E}_{\boldsymbol{\mu}, \boldsymbol{g}_2} \left[ Q\left(\frac{\boldsymbol{\mu}^\top(\boldsymbol{\mu} + \frac{\sigma}{\sqrt{m}} \boldsymbol{g}_2)}{\sigma \left\|\boldsymbol{\mu} + \frac{\sigma}{\sqrt{m}} \boldsymbol{g}_2\right\|_{\ell_2}}\right) \right] \\
&= \mathbb{E}_{\boldsymbol{\mu}, \boldsymbol{g}_2} \left[ Q\left(\frac{1 + \frac{\sigma}{\sqrt{m}} \boldsymbol{\mu}^\top \boldsymbol{g}_2}{\sigma \sqrt{1 + 2\frac{\sigma}{\sqrt{m}} \boldsymbol{\mu}^\top \boldsymbol{g}_2 + \frac{\sigma^2}{m} \|\boldsymbol{g}_2\|_{\ell_2}^2}}\right) \right].
\end{aligned}
$$

Note that for any $\boldsymbol{\mu}$ with $\|\boldsymbol{\mu}\|_{\ell_2} = 1$, we have $\boldsymbol{\mu}^\top \boldsymbol{g}_2 \sim \mathcal{N}(0, 1)$. Therefore, we can write

$$\boldsymbol{\mu}^\top \boldsymbol{g}_2 = g \quad \text{where} \quad g \sim \mathcal{N}(0, 1),$$

and let $\boldsymbol{U} \in \mathbb{R}^{d \times d}$ be a unitary matrix with first row being $\boldsymbol{\mu}$. We can write

$$\|\boldsymbol{g}_2\|_{\ell_2}^2 = \|\boldsymbol{U}\boldsymbol{g}_2\|_{\ell_2}^2 = g^2 + h \quad \text{where} \quad h \sim \mathcal{X}_{d-1}^2.$$

Here, $\mathcal{X}_{d-1}^2$ denotes chi-squared distribution with $(d-1)$ degrees of freedom. Then, we get

$$
\begin{aligned}
\mathbb{P}(y^\star_{\text{att-1}}(\boldsymbol{Z}) \neq y) &= \mathbb{E}_{g,h} \left[ Q\left(\frac{1 + \frac{\sigma}{\sqrt{m}} g}{\sigma \sqrt{1 + 2\frac{\sigma}{\sqrt{m}} g + \frac{\sigma^2}{m}(g^2 + h)}}\right) \right] \\
&= \mathbb{E}_{g,h} \left[ Q\left(\frac{1 + \frac{\sigma}{\sqrt{m}} g}{\sigma \sqrt{(1 + \frac{\sigma}{\sqrt{m}} g)^2 + \frac{\sigma^2}{m} h}}\right) \right], \\
&= \mathbb{E}_{g,h} \left[ Q\left(\frac{1 + \varepsilon_\sigma g}{\sigma \sqrt{(1 + \varepsilon_\sigma g)^2 + \varepsilon_\sigma^2 h}}\right) \right],
\end{aligned}
$$

where $\varepsilon_\sigma := \sigma / \sqrt{m}$. It completes the proof of (9).

Next, we derive an upper bound for $\mathbb{P}(y^\star_{\text{att-1}}(\mathbf{Z}) \neq y)$. Let $c := \varepsilon_\sigma^{-1}$. Then we have

$$
\mathbb{P}(y^\star_{\text{att-1}}(\mathbf{Z}) \neq y) = \mathbb{E}_{g,h}\left[Q\left(\frac{c+g}{\sigma\sqrt{(c+g)^2+h}}\right)\right]
$$

$$
= \mathbb{E}_{g\geq-\frac{c}{2},h}\left[Q\left(\frac{c+g}{\sigma\sqrt{(c+g)^2+h}}\right)\right] + \mathbb{E}_{g<-\frac{c}{2},h}\left[Q\left(\frac{c+g}{\sigma\sqrt{(c+g)^2+h}}\right)\right]
$$

$$
\leq \mathbb{E}_{g\geq-\frac{c}{2},h}\left[Q\left(\frac{c+g}{\sigma\sqrt{(c+g)^2+h}}\right)\right] + Q(c/2)
$$

$$
= \mathbb{E}_{g\geq-\frac{c}{2},h}\left[Q\left(\frac{1}{\sigma\sqrt{1+h/(c+g)^2}}\right)\right] + Q(c/2), \tag{24}
$$

where the inequality comes from the fact that $\mathbb{P}(g \leq -c/2) = Q(c/2)$ and $Q(x) \leq 1$ for any $x \in \mathbb{R}$. Next, we have

$$
\frac{1}{\sqrt{1+h/(c+g)^2}} \geq 1 - \frac{1}{2}\frac{h}{(c+g)^2} \geq 1 - \frac{2h}{c^2}.
$$

Here the first inequality comes from that $\frac{1}{\sqrt{1+x}} \geq 1 - \frac{1}{2}x$ and the second utilizes that $g \geq -\frac{c}{2}$.

Since $h \sim \mathcal{X}^2_{d-1}$, from the Laurent-Massart inequality (Laurent & Massart, 2000), we have that

$$
\mathbb{P}\left(h \geq d - 1 + 2\sqrt{(d-1)t_1} + 2t_1\right) \leq e^{-t_1}.
$$

Therefore, we have that with probability at least $1 - e^{-t_1}$

$$
\frac{1}{\sqrt{1+h/(c+g)^2}} \geq 1 - \frac{2(d-1+2\sqrt{(d-1)t_1}+2t_1)}{c^2}.
$$

Setting $t_1 = d$, we get with probability at least $1 - e^{-d}$

$$
\frac{1}{\sqrt{1+h/(c+g)^2}} \geq 1 - \frac{10d}{c^2}.
$$

Combining the result with (24), since $Q(x) \leq 1$ for $x \in \mathbb{R}$ and $Q(x) \leq e^{-x^2/2}$ for $x > 1$, we get that

$$
\mathbb{P}(y^\star_{\text{att-1}}(\mathbf{Z}) \neq y) \leq e^{-d} + Q(c/2) + Q\left(\frac{1}{\sigma}\left(1 - \frac{10d}{c^2}\right)\right)
$$

$$
\leq e^{-d} + e^{-1/8\varepsilon_\sigma^2} + Q\left(\frac{1}{\sigma}\left(1 - 10d\varepsilon_\sigma^2\right)\right).
$$

It completes the proof. ∎

# C   Analysis of Multi-layer Linear Attention

## C.1   Proof of Proposition 1

**Proof.**  We consider the following model constructions for the attention matrices in the $\ell$th layer, $\ell \in [L]$ and the final linear prediction head:

$$
\ell\text{th layer:}\quad \mathbf{W}_{q\ell}\mathbf{W}_{k\ell}^\top = \begin{bmatrix} \mathbf{I}_d & 0 \\ 0 & 0 \end{bmatrix} \quad \text{and} \quad \mathbf{W}_{v\ell} = \begin{bmatrix} a_\ell\mathbf{I}_d & 0 \\ 0 & b_\ell \end{bmatrix};
$$

$$
\text{Prediction head:}\quad \mathbf{h} = \begin{bmatrix} \mathbf{0}_d \\ c \end{bmatrix}. \tag{25}
$$

Suppose the input to $\ell$th layer is

$$
\mathbf{Z}_\ell = \begin{bmatrix} \mathbf{X}_\ell & \mathbf{y}_\ell \\ \mathbf{x}_\ell^\top & y_\ell \end{bmatrix} \in \mathbb{R}^{(n+1)\times(d+1)} \quad \text{where} \quad \mathbf{Z}_1 = \mathbf{Z} = \begin{bmatrix} \mathbf{X} & \mathbf{y} \\ \mathbf{x}^\top & 0 \end{bmatrix}.
$$

Recapping the model construction from (25), the $\ell$th layer output returns

$$\left(Z_\ell W_{q\ell} W_{k\ell}^\top Z_\ell^\top M\right) Z_\ell W_{v\ell} = \begin{bmatrix} X_\ell & y_\ell \\ x_\ell^\top & y_\ell \end{bmatrix} \begin{bmatrix} I_d & 0 \\ 0 & 0 \end{bmatrix} \begin{bmatrix} X_\ell^\top & x_\ell \\ y_\ell^\top & y_\ell \end{bmatrix} M \begin{bmatrix} X_\ell & y_\ell \\ x_\ell^\top & y_\ell \end{bmatrix} \begin{bmatrix} a_\ell I_d & 0 \\ 0 & b_\ell \end{bmatrix}$$

$$= \begin{bmatrix} X_\ell X_\ell^\top & X_\ell x_\ell \\ x_\ell^\top X_\ell^\top & x_\ell^\top x_\ell \end{bmatrix} \begin{bmatrix} I_n & 0 \\ 0 & 0 \end{bmatrix} \begin{bmatrix} a_\ell X_\ell & b_\ell y_\ell \\ a_\ell x_\ell^\top & b_\ell y_\ell \end{bmatrix}$$

$$= \begin{bmatrix} a_\ell X_\ell X_\ell^\top X_\ell & b_\ell X_\ell X_\ell^\top y_\ell \\ a_\ell x_\ell^\top X_\ell^\top X_\ell & b_\ell x_\ell^\top X_\ell^\top y_\ell \end{bmatrix}. \tag{26}$$

Therefore, following (5), the input of $(\ell + 1)$th layer is

$$Z_{\ell+1} = Z_\ell + \begin{bmatrix} a_\ell X_\ell X_\ell^\top X_\ell & b_\ell X_\ell X_\ell^\top y_\ell \\ a_\ell x_\ell^\top X_\ell^\top X_\ell & b_\ell x_\ell^\top X_\ell^\top y_\ell \end{bmatrix}$$

$$= \begin{bmatrix} X_\ell + a_\ell X_\ell X_\ell^\top X_\ell & y_\ell + b_\ell X_\ell X_\ell^\top y_\ell \\ x_\ell^\top + a_\ell x_\ell^\top X_\ell^\top X_\ell & y_\ell + b_\ell x_\ell^\top X_\ell^\top y_\ell \end{bmatrix} \in \mathbb{R}^{(n+1)\times(d+1)}. \tag{27}$$

- **Label propagation:** We first focus on deriving label propagation results. Suppose that we have

$$a_\ell = 0 \quad \text{for} \quad \ell \in [L].$$

Then following (26), the output of $\ell$'th layer takes the following form:

$$\left(Z_\ell W_{q\ell} W_{k\ell}^\top Z_\ell^\top M\right) Z_\ell W_{v\ell} = \begin{bmatrix} 0 & b_\ell X_\ell X_\ell^\top y_\ell \\ 0 & b_\ell x_\ell^\top X_\ell^\top y_\ell \end{bmatrix}.$$

Here, the first $d$ coordinates of each token's output are zeros, and therefore, the corresponding input coordinates remain unchanged, and we have

$$X_\ell \equiv X \quad \text{and} \quad x_\ell \equiv x \quad \text{for} \quad \ell \in [L].$$

The prediction (based on the last token output and after applying prediction head) is given by

$$f_{\text{all-}L}(Z) = c b_L x^\top X^\top y_L. \tag{28}$$

We next focus on obtaining $y_L$. From (27), we have

$$y_{\ell+1} = y_\ell + b_\ell X X^\top y_\ell = (I + b_\ell X X^\top) y_\ell.$$

Therefore,

$$y_L = \prod_{\ell=1}^{L-1} (I + b_\ell X X^\top) y.$$

Combining with (28) results in

$$f_{\text{all-}L}(Z) = c b_L x^\top X^\top \prod_{\ell=1}^{L-1} (I + b_\ell X X^\top) y = c b_L x^\top \prod_{\ell=1}^{L-1} (I + b_\ell X^\top X) X^\top y.$$

It completes the proof.

- **Feature propagation:** We now focus on the feature propagation setting. In contrast to the label propagation, let us assume that

$$a_\ell \to \infty \quad \text{and} \quad b_\ell \to 0^+ \quad \text{for} \quad \ell \in [L].$$

The prediction (following (26), based on the last token output and after applying prediction head) is given by

$$f_{\text{all-}L}(Z) = c b_L x_L^\top X_L^\top y_L. \tag{29}$$

We first obtain $y_L$. From (27) (since $b_\ell \to 0$), we have

$$y_{\ell+1} = y_\ell + b_\ell X X^\top y_\ell = y_\ell.$$

Therefore,

$$\boldsymbol{y}_\ell \equiv \boldsymbol{y} \quad \text{for} \quad \ell \in [L].$$

Next, we focus on $\boldsymbol{X}_L, \boldsymbol{x}_L$. From (27), as $a_\ell \to \infty$, we have

$$\boldsymbol{X}_{\ell+1} = \boldsymbol{X}_\ell + a_\ell \boldsymbol{X}_\ell \boldsymbol{X}_\ell^\top \boldsymbol{X}_\ell = \boldsymbol{X}_\ell(\boldsymbol{I} + a_\ell \boldsymbol{X}_\ell^\top \boldsymbol{X}_\ell) = a_\ell \boldsymbol{X}_\ell \boldsymbol{X}_\ell^\top \boldsymbol{X}_\ell;$$
$$\boldsymbol{x}_{\ell+1}^\top = \boldsymbol{x}_\ell^\top + a_\ell \boldsymbol{x}_\ell^\top \boldsymbol{X}_\ell^\top \boldsymbol{X}_\ell = \boldsymbol{x}_\ell^\top(\boldsymbol{I} + a_\ell \boldsymbol{X}_\ell^\top \boldsymbol{X}_\ell) = a_\ell \boldsymbol{x}_\ell^\top \boldsymbol{X}_\ell^\top \boldsymbol{X}_\ell.$$

Therefore,

$$
\begin{aligned}
\boldsymbol{X}_L &= a_{L-1} \boldsymbol{X}_{L-1}(\boldsymbol{X}_{L-1}^\top \boldsymbol{X}_{L-1}) \\
&= a_{L-1} a_{L-2}^3 \boldsymbol{X}_{L-2}(\boldsymbol{X}_{L-2}^\top \boldsymbol{X}_{L-2})^{\frac{3^2-1}{2}} \\
&= a_{L-1} a_{L-2}^3 a_{L-3}^{3^2} \boldsymbol{X}_{L-3}(\boldsymbol{X}_{L-3}^\top \boldsymbol{X}_{L-3})^{\frac{3^3-1}{2}} \\
&= \cdots \\
&= a_{L-1} a_{L-2}^3 a_{L-3}^{3^2} ... a_1^{3^{L-2}} \boldsymbol{X}(\boldsymbol{X}^\top \boldsymbol{X})^{\frac{3^{L-1}-1}{2}},
\end{aligned}
$$

and

$$
\begin{aligned}
\boldsymbol{x}_L^\top &= a_{L-1} \boldsymbol{x}_{L-1}^\top(\boldsymbol{X}_{L-1}^\top \boldsymbol{X}_{L-1}) \\
&= a_{L-1} a_{L-2}^3 \boldsymbol{x}_{L-2}^\top(\boldsymbol{X}_{L-2}^\top \boldsymbol{X}_{L-2})^{\frac{3^2-1}{2}} \\
&= a_{L-1} a_{L-2}^3 a_{L-3}^{3^2} \boldsymbol{x}_{L-3}^\top(\boldsymbol{X}_{L-3}^\top \boldsymbol{X}_{L-3})^{\frac{3^3-1}{2}} \\
&= \cdots \\
&= a_{L-1} a_{L-2}^3 a_{L-3}^{3^2} ... a_1^{3^{L-2}} \boldsymbol{x}^\top(\boldsymbol{X}^\top \boldsymbol{X})^{\frac{3^{L-1}-1}{2}}.
\end{aligned}
$$

Combining all together with (29), we have that

$$
\begin{aligned}
f_{\text{all-}L}(\boldsymbol{Z}) &= c b_L \boldsymbol{x}_L^\top \boldsymbol{X}_L^\top \boldsymbol{y}_L \\
&= c b_L \left(\prod_{\ell=1}^{L-1} a_\ell^{3^{L-1-\ell}}\right)^2 \boldsymbol{x}^\top(\boldsymbol{X}^\top \boldsymbol{X})^{3^{L-1}-1} \boldsymbol{X}^\top \boldsymbol{y}.
\end{aligned}
$$

It completes the proof. ∎

## C.2  Proof of Proposition 2

**Proof.** The proof follows directly by adopting the same model construction and proof strategy as in Proposition 1, under the additional assumption that

$$a_\ell = a \quad \text{and} \quad b_\ell = b \quad \text{for} \quad \ell \in [L].$$

∎

## C.3  Proof of Lemma 1

**Proof.** In the proof of Proposition 1, we showed how to derive the label and feature propagation results by restricting the construction to either $a_\ell \equiv 0$ (for label propagation) or $(a_\ell \to \infty, b_\ell \to 0)$ (for feature propagation). Here, we consider a propagation process without imposing restrictions on the choices of $(a_\ell, b_\ell)$, and study the form of the final prediction returned by the model.

To avoid the notation conflict, we express the matrix $\boldsymbol{A}$ in (12) as

$$\boldsymbol{A} = \sum_{k=0}^{K} e_k(\boldsymbol{X}^\top \boldsymbol{X})^k$$

and let $\boldsymbol{e} = [e_0 \ e_2 \ \cdots \ e_{(3^L-3)/2}]^\top \in \mathbb{R}^{K+1}$.

Recall the same model construction used in the proof of Proposition 1, defined in (25). From (26), we have that

$$f_{\text{att-}L}(\mathbf{Z}) = cb_L \mathbf{x}_L^\top \mathbf{X}_L^\top \mathbf{y}_L$$

where following (27), we have

$$\mathbf{X}_{\ell+1} = \mathbf{X}_\ell(\mathbf{I} + a_\ell \mathbf{X}_\ell^\top \mathbf{X}_\ell),$$
$$\mathbf{x}_{\ell+1}^\top = \mathbf{x}_\ell^\top(\mathbf{I} + a_\ell \mathbf{X}_\ell^\top \mathbf{X}_\ell),$$
$$\mathbf{y}_{\ell+1} = (\mathbf{I} + b_\ell \mathbf{X}_\ell \mathbf{X}_\ell^\top)\mathbf{y}_\ell.$$

At each layer, the operations performed are linear combinations and multiplications involving $\mathbf{X}_\ell^\top \mathbf{X}_\ell$ and identity matrices scaled by the parameters $(a_\ell, b_\ell)$. Thus, each coefficient $e_k$ of $(\mathbf{X}^\top \mathbf{X})^k$ depends smoothly on the scalar parameters $(a_\ell, b_\ell)$.

From (26) and (27), we have that

$$\begin{aligned}
f_{\text{att-}L}(\mathbf{Z}) &= cb_L \mathbf{x}_L^\top \mathbf{X}_L^\top \mathbf{y}_L \\
&= cb_L \cdot \mathbf{x}_{L-1}^\top (\mathbf{I} + a_{L-1}\mathbf{X}_{L-1}^\top \mathbf{X}_{L-1})^2 (\mathbf{I} + b_{L-1}\mathbf{X}_{L-1}^\top \mathbf{X}_{L-1})\mathbf{X}_{L-1}^\top \mathbf{y}_{L-1} \\
&= \cdots
\end{aligned} \tag{30}$$

That is, in the final $f_{\text{att-}L}(\mathbf{Z})$ expression, the coefficients corresponding to different degrees of $(\mathbf{X}^\top \mathbf{X})^k$ depend on the model parameters $cb_L$ and $(a_\ell, b_\ell)_{\ell=1}^{L-1}$, which together have at most $2L - 1$ degrees of freedom. Let $\mathbf{c} = [cb_L \; a_1 \; \cdots \; a_{L-1} \; b_1 \; \cdots \; b_{L-1}]^\top$. This means there exists a smooth function $g : \mathbb{R}^{2L-1} \to \mathbb{R}^K$ such that: $\mathbf{e} = g(\mathbf{c})$.

It remains to show that an $L$-layer linear attention model can produce terms involving powers of $\mathbf{X}^\top \mathbf{X}$ up to degree $(3^L - 3)/2$.

Let $f(\mathbf{Z})$ be a function that contains terms of the form $\mathbf{x}^\top(\mathbf{X}^\top \mathbf{X})^k \mathbf{X}^\top \mathbf{y}$ for various powers $k$. Define $\mathcal{P}(f(\mathbf{Z}))$ as the projection that extracts the highest degree $k$ present in $f(\mathbf{Z})$. For example, $\mathcal{P}(\mathbf{x}^\top(\mathbf{I} + (\mathbf{X}^\top \mathbf{X})^2)\mathbf{X}^\top \mathbf{y}) = 2$. Then from (30), we have

$$\begin{aligned}
\mathcal{P}(f_{\text{att-}L}(\mathbf{Z})) &= \mathcal{P}(\mathbf{x}_L^\top \mathbf{X}_L^\top \mathbf{y}_L) \\
&= \mathcal{P}(\mathbf{x}_{L-1}^\top (\mathbf{X}_{L-1}^\top \mathbf{X}_{L-1})^3 \mathbf{X}_{L-1}^\top \mathbf{y}_{L-1}) \\
&= \mathcal{P}(\mathbf{x}_{L-2}^\top (\mathbf{X}_{L-2}^\top \mathbf{X}_{L-2})(\mathbf{X}_{L-2}^\top \mathbf{X}_{L-2})^{3^2}(\mathbf{X}_{L-2}^\top \mathbf{X}_{L-2})^2 \mathbf{X}_{L-2}^\top \mathbf{y}_{L-2}) \\
&= \mathcal{P}(\mathbf{x}_{L-2}^\top (\mathbf{X}_{L-2}^\top \mathbf{X}_{L-2})^{3^2+3} \mathbf{X}_{L-2}^\top \mathbf{y}_{L-2}) \\
&= \mathcal{P}(\mathbf{x}_{L-3}^\top (\mathbf{X}_{L-3}^\top \mathbf{X}_{L-3})(\mathbf{X}_{L-3}^\top \mathbf{X}_{L-3})^{3^3+3^2}(\mathbf{X}_{L-3}^\top \mathbf{X}_{L-3})^2 \mathbf{X}_{L-3}^\top \mathbf{y}_{L-3}) \\
&= \mathcal{P}(\mathbf{x}_{L-3}^\top (\mathbf{X}_{L-3}^\top \mathbf{X}_{L-3})^{3^3+3^2+3} \mathbf{X}_{L-3}^\top \mathbf{y}_{L-3}) \\
&= \cdots \\
&= \mathcal{P}(\mathbf{x}^\top (\mathbf{X}^\top \mathbf{X})^{3^{L-1}+\cdots+3^2+3} \mathbf{X}^\top \mathbf{y}) \\
&= 3^{L-1} + \cdots + 3^2 + 3 = \frac{3^L - 3}{2}.
\end{aligned}$$

It completes the proof.

∎

### C.4  Proof of Theorem 2

**Proof.** Let $\boldsymbol{\xi} \sim \mathcal{N}(0, \mathbf{I})$ and rewrite $y\mathbf{x} = \boldsymbol{\mu} + \sigma\boldsymbol{\xi}$. For any matrix $\mathbf{A} \in \mathbb{R}^{d \times d}$, the prediction error of $\hat{y}_A = \text{sgn}(\mathbf{x}^\top \mathbf{A}\hat{\boldsymbol{\mu}}_s)$ given $\hat{\boldsymbol{\mu}}_s$ returns

$$\begin{aligned}
\mathbb{P}(\hat{y}_A \neq y \mid \hat{\boldsymbol{\mu}}_s) &= \mathbb{P}(y\mathbf{x}^\top \mathbf{A}\hat{\boldsymbol{\mu}}_s < 0 \mid \hat{\boldsymbol{\mu}}_s) \\
&= \mathbb{P}((\boldsymbol{\mu} + \sigma\boldsymbol{\xi})^\top \mathbf{A}\hat{\boldsymbol{\mu}}_s < 0 \mid \hat{\boldsymbol{\mu}}_s) \\
&= Q\left(\frac{\boldsymbol{\mu}^\top \mathbf{A}\hat{\boldsymbol{\mu}}_s}{\sigma \|\mathbf{A}\hat{\boldsymbol{\mu}}_s\|_{\ell_2}}\right).
\end{aligned} \tag{31}$$

For any $A \in \mathbb{R}^{d \times d}$, we can decompose it as

$$A = \sum_{i=1}^{d} \lambda_i u_i v_i^\top$$

where $u_1 = \mu$, $\|u_i\|_{\ell_2} = 1$ and $u_i^\top u_j = 0$ for any $i \neq j$. Let $\lambda_1 > 0$. Then, we get

$$
\begin{aligned}
\mu^\top A \hat{\mu}_s &= \mu^\top (\sum_{i=1}^{d} \lambda_i u_i v_i^\top) \hat{\mu}_s \\
&= \sum_{i=1}^{d} \lambda_i \mu^\top u_i v_i^\top \hat{\mu}_s \\
&= \lambda_1 \mu^\top u_1 v_1^\top \hat{\mu}_s \\
&= \lambda_1 v_1^\top \hat{\mu}_s.
\end{aligned}
\tag{32}
$$

Now consider $\|A\hat{\mu}_s\|_{\ell_2}$ where we have

$$
\begin{aligned}
A\hat{\mu}_s &= \sum_{i=1}^{d} \lambda_i u_i v_i^\top \hat{\mu}_s \\
&= \lambda_1 \mu v_1^\top \hat{\mu}_s + \sum_{i=2}^{d} \lambda_i u_i v_i^\top \hat{\mu}_s.
\end{aligned}
$$

Since $u_i$, $i \neq 1$ is orthogonal to $\mu$, $\lambda_1 \mu v_1^\top \hat{\mu}_s$ is orthogonal to $\sum_{i=2}^{d} \lambda_i u_i v_i^\top \hat{\mu}_s$. Therefore, given $\|u_i\|_{\ell_2} = 1$ for all $i \in [d]$, it obeys

$$\|A\hat{\mu}_s\|_{\ell_2}^2 = \left\| \lambda_1 \mu v_1^\top \hat{\mu}_s \right\|_{\ell_2}^2 + \sum_{i=2}^{d} \left\| \lambda_i u_i v_i^\top \hat{\mu}_s \right\|_{\ell_2}^2 = (\lambda_1 v_1^\top \hat{\mu}_s)^2 + \lambda_1^2 \sum_{i=2}^{d} (\lambda_1^{-1} \lambda_i v_i^\top \hat{\mu}_s)^2. \tag{33}$$

For simplicity, define

$$\Delta(\hat{\mu}_s) = \sum_{i=2}^{d} (\lambda_1^{-1} \lambda_i v_i^\top \hat{\mu}_s)^2$$

where $\Delta(\cdot)$ is a function of $\lambda_1$ and $(\lambda_i, v_i)$'s for $i \geq 2$, and we have

$$\Delta(\hat{\mu}_s) \geq 0 \quad \text{and} \quad \Delta(-\hat{\mu}_s) = \Delta(\hat{\mu}_s).$$

Recall that $\hat{\mu}_s$ is the SPI estimator (cf. (SPI)). Let $|\mathcal{I}| = m$. We can write $\hat{\mu}_s = \mu + \xi'/\sqrt{m}$ where $\xi' \sim \mathcal{N}(0, \sigma^2 I)$.

Using (31), (32) and (33), the classification error becomes

$$
\begin{aligned}
\mathbb{P}(\hat{y}_A \neq y) &= \mathbb{E}_{\hat{\mu}_s} \left[ Q\left( \frac{\mu^\top A \hat{\mu}_s}{\sigma \|A\hat{\mu}_s\|_{\ell_2}} \right) \right] \\
&= \mathbb{E}_{\hat{\mu}_s} \left[ Q\left( \frac{v_1^\top \hat{\mu}_s}{\sigma \sqrt{(v_1^\top \hat{\mu}_s)^2 + \Delta(\hat{\mu}_s)}} \right) \right] \\
&= \mathbb{E}_{v_1^\top \hat{\mu}_s < 0} \left[ Q\left( \frac{v_1^\top \hat{\mu}_s}{\sigma \sqrt{(v_1^\top \hat{\mu}_s)^2 + \Delta(\hat{\mu}_s)}} \right) \right] + \mathbb{E}_{v_1^\top \hat{\mu}_s \geq 0} \left[ Q\left( \frac{v_1^\top \hat{\mu}_s}{\sigma \sqrt{(v_1^\top \hat{\mu}_s)^2 + \Delta(\hat{\mu}_s)}} \right) \right].
\end{aligned}
$$

First, note that for any $x > 0$, $Q(x) < 0.5 < Q(-x)$. Therefore, the optimal choice of $v_1 \in \mathbb{R}^d$ that minimizes $\mathbb{P}(\hat{y}_A \neq y)$ is contained within the set of $v_1$ values that maximize $\mathbb{P}(v_1^\top \hat{\mu}_s > 0)$. Let $v_1^\star := \arg\max_{v_1 \in \mathbb{R}^d} \mathbb{P}(v_1^\top \hat{\mu}_s > 0)$. Given that $\hat{\mu}_s \sim \mathcal{N}(\mu, \sigma^2/m I)$, we have that $v_1^\star = c\mu$ for $c > 0$. Let $c = 1$ and therefore, $v_1^\star = \mu$ without loss of generality (since $\lambda_1$ can be any positive scalar). Then we obtain

$$\min_{A \in \mathbb{R}^{d \times d}} \mathbb{P}(\hat{y}_A \neq y) = \min_{\Delta} \mathbb{E}_{\hat{\mu}_s} \left[ Q\left( \frac{\mu^\top \hat{\mu}_s}{\sigma \sqrt{(\mu^\top \hat{\mu}_s)^2 + \Delta(\hat{\mu}_s)}} \right) \right].$$

Let $f(\hat{\mu}_s)$ be the probability density function of $\hat{\mu}_s$. Since $\hat{\mu}_s \sim \mathcal{N}(\mu, \sigma^2/m\mathbf{I})$, then it satisfies

$$f(\hat{\mu}_s) \geq f(-\hat{\mu}_s) \quad \text{for any } \mu^\top \hat{\mu}_s > 0. \tag{34}$$

Therefore, the classification error becomes

$$\mathbb{P}(\hat{y}_A \neq y \mid v_1 = \mu) = \int_{\hat{\mu}_s} f(\hat{\mu}_s) Q\left(\frac{\mu^\top \hat{\mu}_s}{\sigma \sqrt{(\mu^\top \hat{\mu}_s)^2 + \Delta(\hat{\mu}_s)}}\right) d\hat{\mu}_s$$

$$= \int_{\mu^\top \hat{\mu}_s > 0} f(\hat{\mu}_s) Q\left(\frac{\mu^\top \hat{\mu}_s}{\sigma \sqrt{(\mu^\top \hat{\mu}_s)^2 + \Delta(\hat{\mu}_s)}}\right) + f(-\hat{\mu}_s) Q\left(\frac{-\mu^\top \hat{\mu}_s}{\sigma \sqrt{(\mu^\top \hat{\mu}_s)^2 + \Delta(\hat{\mu}_s)}}\right) d\hat{\mu}_s$$

$$= \int_{\mu^\top \hat{\mu}_s > 0} (f(\hat{\mu}_s) - f(-\hat{\mu}_s)) \, Q\left(\frac{\mu^\top \hat{\mu}_s}{\sigma \sqrt{(\mu^\top \hat{\mu}_s)^2 + \Delta(\hat{\mu}_s)}}\right) + f(-\hat{\mu}_s) d\hat{\mu}_s.$$

Following (34), to minimize the error, we need minimize $Q\left(\frac{\mu^\top \hat{\mu}_s}{\sigma \sqrt{(\mu^\top \hat{\mu}_s)^2 + \Delta(\hat{\mu}_s)}}\right)$ for $\mu^\top \hat{\mu}_s > 0$, which can be easily done by choosing $\lambda_i = 0$ for $i \geq 2$. Then we get $\Delta(\hat{\mu}_s) \equiv 0$. Therefore, the optimal solution set $\mathcal{A}^\star$ defined in Theorem 2 satisfies:

$$\mathcal{A}^\star = \left\{ \lambda_1 \mu \mu^\top \mid \lambda_1 > 0 \right\}.$$

Combining all together, we obtain

$$\min_{A \in \mathbb{R}^{d \times d}} \mathbb{P}(\hat{y}_A \neq y) = \int_{\mu^\top \hat{\mu}_s > 0} (f(\hat{\mu}_s) - f(-\hat{\mu})) \, Q\left(\frac{1}{\sigma}\right) + f(-\hat{\mu}_s) d\hat{\mu}_s$$

$$= \int_{\mu^\top \hat{\mu}_s > 0} f(\hat{\mu}_s) d\hat{\mu}_s \cdot Q\left(\frac{1}{\sigma}\right) + \int_{\mu^\top \hat{\mu}_s < 0} f(\hat{\mu}_s) d\hat{\mu}_s \cdot \left(1 - Q\left(\frac{1}{\sigma}\right)\right)$$

$$= Q\left(-\frac{\sqrt{m}}{\sigma}\right) Q\left(\frac{1}{\sigma}\right) + Q\left(\frac{\sqrt{m}}{\sigma}\right)\left(1 - Q\left(\frac{1}{\sigma}\right)\right)$$

$$= \left(1 - Q\left(\frac{\sqrt{m}}{\sigma}\right)\right) Q\left(\frac{1}{\sigma}\right) + Q\left(\frac{\sqrt{m}}{\sigma}\right)\left(1 - Q\left(\frac{1}{\sigma}\right)\right)$$

$$= Q\left(\frac{1}{\sigma}\right) + Q\left(\frac{\sqrt{m}}{\sigma}\right) - 2Q\left(\frac{\sqrt{m}}{\sigma}\right) Q\left(\frac{1}{\sigma}\right).$$

It completes the proof. ∎

## C.5 Proof of Theorem 4

**Proof.** Recap from Proposition 1. For any $L$-layer attention model with $L \geq 2$, it can output

$$f_{\text{att-}L}(\mathbf{Z}) = x^\top (X^\top X/n - \sigma^2 \mathbf{I})\hat{\mu}_s. \tag{35}$$

Let

$$\hat{y} = \text{sgn}(f_{\text{att-}L}(\mathbf{Z}))$$

with $f_{\text{att-}L}(\mathbf{Z})$ defined in (35). Then we have

$$\mathbb{P}(y^\star_{\text{att-}L}(\mathbf{Z}) \neq y) \leq \mathbb{P}(\hat{y} \neq y).$$

Therefore, in the following, we focus on upper-bounding the classification error $\mathbb{P}(\hat{y} \neq y)$ corresponding to (35). Given that the optimal prediction under the form $\text{sgn}(x^\top A\hat{\mu}_s)$ is given by $\hat{y}_{\mu\mu^\top} := \text{sgn}(x^\top \mu \mu^\top \hat{\mu}_s)$ (cf. Theorem 2), with its corresponding error presented in (13). To analyze the performance of $\hat{y}$, we study its difference from the prediction $\hat{y}_{\mu\mu^\top}$.

To begin with, let $g_i = \xi_i/\sigma \sim \mathcal{N}(0, \mathbf{I})$ and $g = \sum_{i=1}^n \xi_i/\sigma \sqrt{n} \sim \mathcal{N}(0, \mathbf{I})$. For simplicity, let $A := X^\top X/n - \sigma^2 \mathbf{I}$. We get

$$A = \frac{1}{n} X^\top X - \sigma^2 \mathbf{I}$$

$$= \frac{1}{n}\left(\sum_{i=1}^n \mu\mu^\top + \mu\xi_i^\top + \xi_i\mu^\top + \xi_i\xi_i^\top\right) - \sigma^2 \mathbf{I}$$

$$= \mu\mu^\top + \frac{\sigma}{\sqrt{n}}(\mu g^\top + g\mu^\top) + \sigma^2\left(\frac{\sum_{i=1}^n g_i g_i^\top}{n} - \mathbf{I}\right).$$

Recall (31) from the proof of Theorem 2. Our goal is to bound

$$\mathbb{P}(\hat{y} \neq y) = \mathbb{E}_{\hat{\mu}}\left[Q\left(\frac{\mu^\top A \hat{\mu}_s}{\sigma \|A\hat{\mu}_s\|_{\ell_2}}\right)\right].$$

Define

$$\Delta := A - \mu\mu^\top = \frac{\sigma}{\sqrt{n}}(\mu g^\top + g\mu^\top) + \sigma^2\left(\frac{\sum_{i=1}^n g_i g_i^\top}{n} - I\right). \tag{36}$$

From the Laurent-Massart inequality (Laurent & Massart, 2000), we have that with probability at least $1 - e^{-t_1}$ (assuming $t_1 \geq d$), the first term of (36) can be bounded by

$$\frac{1}{\sqrt{n}}\left\|\mu g^\top + g\mu^\top\right\| \leq \frac{2\|g\|}{\sqrt{n}} \leq 6\sqrt{\frac{t_1}{n}}. \tag{37}$$

Additionally, from Neopane (2018), we have that with probability at least $1 - e^{-t_2}$ (assuming $t_2 \geq d$), the second term of $\Delta$ (cf. (36)) is bounded by (with a universal constant $C > 0$)

$$\left\|\frac{\sum_{i=1}^n g_i g_i^\top}{n} - I\right\| \leq C \cdot \sqrt{\frac{t_2}{n}}. \tag{38}$$

Combining (37) and (38), we get with probability at least $1 - 2e^{-t}$ (for $t \geq d$)

$$\|\Delta\| \leq C_1 \sqrt{\frac{t}{n}} \quad \text{where} \quad C_1 := 6\sigma + C\sigma^2.$$

We also bound $\|\hat{\mu}_s\|$ as follows. Let $\hat{\mu}_s = \mu + \sigma/\sqrt{m}g' \sim \mathcal{N}(\mu, \sigma^2 m I)$, similar to (37), with probability at least $1 - e^{-t_3}$ (assuming $2d \leq t_3 \leq m/4\sigma^2$), we can bound

$$\|\hat{\mu}_s\| \leq 1 + \frac{\sigma}{\sqrt{m}}\|g'\| \leq 1 + 3\sigma\sqrt{\frac{t_3}{m}} \leq 3.$$

Then consider a significantly large $n$ (to ensure that $\|\Delta\| \leq 1/12$, e.g., $n \geq (12C_1)^2 t$). With probability at least $1 - 3e^{-\min(t,t_3)}$ and suppose that $\mu^\top \hat{\mu}_s > 0.5$, we can bound

$$\left|\frac{\mu^\top A \hat{\mu}_s}{\|A\hat{\mu}_s\|_{\ell_2}} - \frac{\mu^\top \mu\mu^\top \hat{\mu}_s}{\|\mu\mu^\top \hat{\mu}_s\|_{\ell_2}}\right| = \left|\frac{\mu^\top(\Delta + \mu\mu^\top)\hat{\mu}_s}{\|(\Delta + \mu\mu^\top)\hat{\mu}_s\|_{\ell_2}} - \frac{\mu^\top \mu\mu^\top \hat{\mu}_s}{\|\mu\mu^\top \hat{\mu}_s\|_{\ell_2}}\right|$$

$$\leq \left|\frac{\mu^\top \Delta \hat{\mu}_s}{\min(\|(\Delta + \mu\mu^\top)\hat{\mu}_s\|_{\ell_2}, \|\mu\mu^\top \hat{\mu}_s\|_{\ell_2})}\right|$$

$$\leq \frac{\|\Delta\| \cdot \|\hat{\mu}_s\|}{\mu^\top \hat{\mu}_s - \|\Delta\| \cdot \|\hat{\mu}_s\|}$$

$$\leq 4\|\Delta\| \cdot \|\hat{\mu}_s\|$$

$$\leq C_2 \sqrt{\frac{t}{n}} \quad \text{where} \quad C_2 := 12C_1.$$

Now, we are ready to bound the classification error, where we get

$$\mathbb{P}(\hat{y} \neq y) = \mathbb{E}_{\hat{\mu}}\left[Q\left(\frac{\mu^\top A \hat{\mu}_s}{\sigma \|A\hat{\mu}_s\|_{\ell_2}}\right)\right]$$

$$= \mathbb{E}_{\hat{\mu}}\left[Q\left(\frac{1}{\sigma} + \frac{1}{\sigma}\left(\frac{\mu^\top A \hat{\mu}_s}{\|A\hat{\mu}_s\|_{\ell_2}} - \frac{\mu^\top \mu\mu^\top \hat{\mu}_s}{\|\mu\mu^\top \hat{\mu}_s\|_{\ell_2}}\right)\right)\right]$$

$$\leq \mathbb{P}(\mu^\top \hat{\mu}_s > 0.5)\left(Q\left(\frac{1 - C_2\sqrt{t/n}}{\sigma}\right) + 3e^{-\min(t,t_3)}\right) + \mathbb{P}(\mu^\top \hat{\mu}_s < 0.5)$$

$$\leq Q\left(\frac{1 - C_2\sqrt{t/n}}{\sigma}\right) + 3e^{-\min(t,t_3)} + Q\left(\frac{\sqrt{m}}{2\sigma}\right).$$

Choosing $t = t_3 = 2d$, since $m/4\sigma^2 \geq 2d$, we obtain

$$\mathbb{P}\left(\hat{y} \neq y\right) \leq Q\left(\frac{1 - C_2 \sqrt{2d/n}}{\sigma}\right) + 3e^{-2d} + 0.5e^{-d}$$

$$\leq Q\left(\frac{1 - C_2 \sqrt{2d/n}}{\sigma}\right) + e^{-d}.$$

It completes the proof. ■

