# OpenReview forum: "When and How Unlabeled Data Provably Improve In-Context Learning"
_NeurIPS.cc/2025/Conference — NeurIPS 2025 poster_

### Official Review · Reviewer_qhJr · 2025-06-23

**Clarity:** 2
**Significance:** 3
**Originality:** 4
**Rating:** 4
**Confidence:** 2

**Summary:**

This paper investigates how transformer-based models perform ICL in a semi-supervised setting where some demonstration labels are missing. The authors focus on analyzing the effectiveness of linear attention transformers at different depths. It finds that single-layer linear attention achieves the optimal supervised solution but fails to leverage unlabeled data while multi-layer or loop linear attention can exploit unlabeled data by approximating estimators of the form. Consequently, the paper proposes looping-based methods for tabular foundation models like TabPFN, which significantly improve semi-supervised classification accuracy.

**Questions:**

1.	What is w.p in eq(1).

2.	Eq (3): is Z the demonstration example or instruction without query x or with query x but without x’s label?

3.	Iine 138, why input of layer l only contains attention part without ffn part?

4.	Line 142, I do not understand this question? The prediction of y is decided by res(attnZlast)+ffn(attnZlast+res(Zlast-1)) according to transformer block.

5.	How sensitive is the performance of the looping-based TabPFN method to the accuracy of initial pseudo-labels? Would noisy pseudo-labels mislead the iteration? Does the portion between label, unlabel, psedu-label sensitive?

6.	How should one determine the optimal number of loops in practice? Does it correlate with data properties like noise level or number of unlabeled samples?

**Ethical Concerns:**

["NO or VERY MINOR ethics concerns only"]

**Final Justification:**

Author's clarification resolved my questions. I keep my positive review for this paper.

**Limitations:**

yes

**Quality:**

3

**Strengths And Weaknesses:**

Strengths

1.	Theoretical Contribution: Provides the comprehensive theoretical characterization of semi-supervised in-context learning (SS-ICL) using linear attention models.

2.	Polynomial Estimator Characterization: Establishes that deeper models can compute higher-order polynomials of the empirical covariance, and thus emulate powerful iterative methods like EM or pseudo-labeling.

3.	Uncovers the limitations of shallow models under isotropic priors, tying it to how expectation washes out task-specific signal.

Weaknesses

1.	Limited to Model structure: theoretical results focus only on linear attention models. While the paper does include empirical comparison to softmax-based Transformers, no theoretical treatment is provided for these more widely used architectures.

2.	Assumption of Isotropic Priors: The analysis assumes task means are uniformly sampled from the unit sphere, which may not reflect realistic downstream data distributions.

3.	While effective, the looping approach introduces additional compute overhead, and the paper does not explore how to optimize this trade-off.

4.	Line 296-298 How to decide α is confused. does it need require domain knowledge or validation data to set optimally?

5.	See Questions.

---

> ### Author Rebuttal · Authors · 2025-07-31
>
> We thank the reviewer for the detailed feedback. Below, we provide our responses to the comments.
>
> > *W1: Limited to Model structure: theoretical results focus only on linear attention models. While the paper does include empirical comparison to softmax-based Transformers, no theoretical treatment is provided for these more widely used architectures.*
>
> **Response:** We kindly refer the reviewer to our **responses to W1 and W3 of Reviewer 9Hjq, and W1 of Reviewer GrWH**  for detailed discussions addressing this concern.
>
> > *W2: Assumption of Isotropic Priors: The analysis assumes task means are uniformly sampled from the unit sphere, which may not reflect realistic downstream data distributions.*
>
> **Response:** We use the isotropic GMM to ensure theoretical tractability and to investigate the effects of model depth and unlabeled data in a controlled setting. While it may not capture the full complexity of real-world data, it serves as a principled and analytically tractable starting point for studying algorithms designed to improve semi-supervised performance.
>
> Building on the core conclusion of our work that deeper or looped models enhance semi-supervised learning, we proposed the Looping TabPFN algorithm, which effectively leverages unlabeled data to improve performance in practical settings. Thus, while our theoretical framework assumes isotropic priors for analytical clarity, the key insights extend to broader scenarios. We agree that further exploration under more realistic or structured priors is valuable and constitutes a promising direction for future work.
>
>
> > *W3: While effective, the looping approach introduces additional compute overhead, and the paper does not explore how to optimize this trade-off.*
>
> **Response:** We agree that the trade-off between accuracy and computational cost is an important consideration. While this work does not focus primarily on optimizing that trade-off, both our synthetic and TabPFN experiments show that only a small number of iterations (typically fewer than 5) are sufficient to achieve the best performance. In fact, the first few iterations contribute the most to performance gains, suggesting that the additional compute overhead is relatively modest given the improvement achieved.
>
> We also refer the reviewer to our **response to Question of Reviewer 9Hjq**, where we further discuss the effectiveness and practicality of the looping strategy.
>
>
> > *W4: Line 296-298 How to decide α is confused. does it need require domain knowledge or validation data to set optimally?*
>
> **Response:** In our SSPI algorithm (Lines 246–247), the parameter $\alpha$ is used under the assumption that its optimal value is known. In our experiments, $\alpha$ is selected based on domain knowledge of the data distribution. However, determining the optimal value of $\alpha$ in practice given only hyperparameters such as the distribution, $d$, $n$, and $p$ is a fundamental mathematical problem. We view this as an important direction for future theoretical investigation.
>
>
> > *Q1: What is w.p in eq(1).*
>
> **Response:** "w.p." means "with probability".
>
> > *Q2: Eq (3): is Z the demonstration example or instruction without query x or with query x but without x’s label?*
>
> **Response:** In Eq (3), $Z$ includes the query $x$ (as the last token in $Z$) but without $x$’s label. This setting is consistent with prior work in the linear attention literature (Ahn et al.(2023), Mahankali et al.(2024), Zhang et al.(2023), Li et al.(2024)).
>
> >*Q3: Iine 138, why input of layer l only contains attention part without ffn part?*
>
> > *Q4: Line 142, I do not understand this question? The prediction of y is decided by res(attnZlast)+ffn(attnZlast+res(Zlast-1)) according to transformer block.*
>
> **Response (Q3&Q4):** In this work, our goal is to analyze the optimization behavior of the attention mechanism, and FFN will introduce additional nonlinearity and complicate theoretical analysis. To that end, we focus only on the attention block, excluding the FFN component and taking the output of the final attention layer as the model output for prediction. This setting is consistent with prior work on multi-layer linear attention (Ahn et al.(2023), Li et al.(2025)). We agree that studying the FFN is also important. As shown in our experiments (Fig. 2(c)), the full Transformer model (attention + MLP) achieves the best performance. However, we believe that a focused study on the attention mechanism, and in particular, the role of depth in attention, remains underexplored and is a necessary step toward a deeper understanding of the full Transformer architecture.
>
> Notably, this restriction applies only to our theoretical study. In our Transformer (Fig 2c, cyan curve) and TabPFN experiments, we do not impose any architectural constraints. This demonstrates that the theory-inspired looping strategy remains effective in practice, even when applied to more complex models.
>
> >*Q5:How sensitive is the performance of the looping-based TabPFN method to the accuracy of initial pseudo-labels? Would noisy pseudo-labels mislead the iteration? Does the portion between label, unlabel, psedu-label sensitive?*
>
> **Response:** Good point. To evaluate this, we examined the performance of TabPFN under varying levels of initial pseudo-label accuracy. The results, shown below, confirm that the quality of pseudo-labeled data plays an important role and higher-quality pseudo-labels lead to greater performance improvements. (Note: Due to time constraints, we conducted the experiment using a single data setting with a fixed labeled set. We randomly sample 100 groups of pseudo-labeled data and report the average relative improvement.)
>
> |Acc of pseudo-labels | 0.2~0.4| 0.4~0.6 | 0.6~0.8 | 0.8~1 |
> |-------------------------|-------------------------|-------------------------|-------------------------|-------------------------|
> |Relative improvement|  5.24% | 6.70% | 7.05% | 10.88% |
>
> Regarding the proportion of labeled, unlabeled, and pseudo-labeled data, our synthetic experiments (e.g., Fig. 1) show that increasing both labeled and unlabeled data consistently improves prediction performance. Motivated by the reviewer's question, we believe that more effective looping strategies could be developed. For example, by gradually assigning pseudo-labels to high-confidence unlabeled examples rather than labeling the entire unlabeled set. Such adaptive approaches may further enhance the performance of the looping method and represent a promising direction for future exploration.
>
> >*Q6: How should one determine the optimal number of loops in practice? Does it correlate with data properties like noise level or number of unlabeled samples?*
>
> **Response:** In our TabPFN experiments, we implement a validation risk criterion to monitor the variance of assigned pseudo-labels (see Lines 710–715 in the Supplementary Material), computed as
> $$Validation~Risk=\frac{1}{n}\sum_i \min(|y^{pseudo}_i-1|,|y^{pseudo}_i+1|).$$
> It is used to determine when to stop looping.
>
> We also evaluated the optimal number of looping iterations with respect to different levels of initial accuracy, and observed an inverse correlation: lower initial accuracy tends to require more loops. However, due to the limited number of datasets and looping iterations, we are cautious about making a definitive conclusion. A more rigorous investigation is needed to establish this connection.
>
> Additionally, we emphasize that based on both our synthetic results (Fig. 1) and TabPFN experiments, the number of loops (or layers) required for strong performance is relatively small (typically fewer than 5) with the initial iterations/layers contributing the most significant improvements.

---

> > ### Comment · Reviewer_qhJr · 2025-08-03
> >
> > Thank you for the clarification and they resolve my questions. I keep my positive review for this paper.

---

### Official Review · Reviewer_HQka · 2025-07-02

**Clarity:** 3
**Significance:** 3
**Originality:** 3
**Rating:** 5
**Confidence:** 3

**Summary:**

This paper studies semi‑supervised in‑context learning, where only some in‑context examples have labels. Under a binary Gaussian‑mixture setting, the authors show that a single‑layer linear‑attention transformer reduces to the supervised plug‑in estimator and therefore ignores unlabeled data. By contrast, deeper or looped linear attention can implement polynomial functions of the sample covariance, effectively performing iterative pseudo‑labeling; only logarithmic depth is required to capture high‑degree terms. They introduce a semi‑supervised plug‑in (SSPI) estimator that blends the supervised estimate with higher‑order covariance corrections and show that multi‑layer attention approaches the optimum as unlabeled samples grow.

**Questions:**

1. Your analysis assumes a shared isotropic task prior for both training and test tasks. How sensitive are the results to this assumption

2. Have you considered tasks drawn from non‑spherical distributions, or mismatched priors between training and inference?

3. In Proposition 1 and Proposition 2, you construct weight matrices to implement polynomial functions of the sample covariance. In practice, are such weights discoverable by gradient‑based training?

4. Can you comment on whether your depth‑induced pseudo‑labeling leads to the kind of “weak‑to‑strong generalization” seen in other semi‑supervised settings, where pseudo‑labels gradually correct errors and expand coverage as more iterations are applied? If so, can the SSPI framework help explain this behavior?

**Ethical Concerns:**

["NO or VERY MINOR ethics concerns only"]

**Final Justification:**

The rebuttal and follow-up discussion addressed my earlier questions on prior misspecification, gradient-based learnability of the constructed estimators, and the connection between depth-induced pseudo-labeling and weak-to-strong generalization. The authors provided thoughtful theoretical clarifications, empirical checks for non-isotropic priors and OOD settings, and a clear mapping of their SSPI framework to observed behavior in iterative pseudo-labeling.

### Resolved issues
- Clarified robustness to non-isotropic priors and OOD settings, with supporting empirical evidence.
- Explained the role and limits of gradient-based training in discovering the constructed weight configurations.
- Linked depth-induced refinement to the weak-to-strong trajectory, with both theoretical and empirical support.

### Remaining considerations
- Analysis is based on an idealized Gaussian mixture setting, and broader applicability to more complex or non-linear regimes is not directly shown.
- Empirical results focus on synthetic and tabular settings; it would be valuable to see extensions to richer domains.

Overall, the paper makes a clear theoretical contribution to understanding semi-supervised ICL, backed by well-motivated experiments. I consider it a solid and relevant contribution to the field.

**Limitations:**

Yes

**Paper Formatting Concerns:**

No concerns

**Quality:**

3

**Strengths And Weaknesses:**

## Strengths

- The paper addresses a realistic and relevant gap. Previous analyses of ICL focused on fully‑supervised tasks; this work is one of the first (to the best of my knowledge) to provide a formal framework for semi‑supervised ICL, where only a subset of demonstrations has labels. This setting arises naturally in many ICL scenarios with scarce labeled data.
- Theorem 1 shows that a one‑layer linear attention model reduces to the supervised plug‑in estimator and thus cannot benefit from unlabeled data. The multi‑layer constructions (Proposition 1, Proposition 2, and Lemma 1) demonstrate that deeper attention can represent polynomial functions of the sample covariance and that only logarithmic depth is needed for high‑degree polynomials.
- Experiments on synthetic mixtures verify that one‑layer attention does not improve with more unlabeled data, whereas adding a few layers substantially boosts accuracy and matches the SSPI predictions. The tabular experiments demonstrate a practical application of looping pseudo‑labeling.
- By relating multi‑layer attention to iterative pseudo‑labeling and EM, the paper theoretically connects ICL and classical semi‑supervised learning, explaining why depth matters.

---

## Weaknesses

- The theoretical analysis relies on an idealized setting: binary Gaussian mixture with isotropic covariance and task means drawn uniformly from the unit sphere. The assumption of a shared task prior across training and test tasks may not hold in more complex scenarios.
- Although the paper shows that deep linear attention can implement polynomial estimators, the constructions rely on hand‑crafted weight matrices rather than learned weights. There is no analysis of whether gradient‑based training can reliably learn these polynomials, nor of how softmax attention or full transformers achieve similar behavior.

---

> ### Author Rebuttal · Authors · 2025-07-31
>
> After reviewing the comments, we suspect that the reviewer may have inadvertently submitted a review targeted at a different paper (also on the topic of ICL),  as the content does not appear to relate to our submission. As a result, we were unable to prepare a rebuttal.
>
>
> If possible, we kindly ask the reviewer to share any key concerns specific to our paper so that we can address them during the discussion period. We look forward to your feedback.

---

> > ### Comment · Reviewer_HQka · 2025-08-01
> > **Updated review submission**
> >
> > I apologize for the earlier submission, I accidentally mixed up my notes. The text below is my corrected review, while the scores remain as originally intended. I realize that some of my questions have already been addressed in your responses to other reviews, but I’m including them here for completeness.
> >
> > ### Summary
> > This paper studies semi‑supervised in‑context learning, where only some in‑context examples have labels.  Under a binary Gaussian‑mixture setting, the authors show that a single‑layer linear‑attention transformer reduces to the supervised plug‑in estimator and therefore ignores unlabeled data.  By contrast, deeper or looped linear attention can implement polynomial functions of the sample covariance, effectively performing iterative pseudo‑labeling; only logarithmic depth is required to capture high‑degree terms.  They introduce a semi‑supervised plug‑in (SSPI) estimator that blends the supervised estimate with higher‑order covariance corrections and show that multi‑layer attention approaches the optimum as unlabeled samples grow.
> >
> > ### Strengths
> > - The paper addresses a realistic and relevant gap. Previous analyses of ICL focused on fully‑supervised tasks; this work is one of the first (to the best of my knowledge) to provide a formal framework for semi‑supervised ICL, where only a subset of demonstrations has labels. This setting arises naturally in many ICL scenarios with scarce labeled data.
> > - Theorem 1 shows that a one‑layer linear attention model reduces to the supervised plug‑in estimator and thus cannot benefit from unlabeled data. The multi‑layer constructions (Proposition 1, Proposition 2, and Lemma 1) demonstrate that deeper attention can represent polynomial functions of the sample covariance and that only logarithmic depth is needed for high‑degree polynomials.
> > - Experiments on synthetic mixtures verify that one‑layer attention does not improve with more unlabeled data, whereas adding a few layers substantially boosts accuracy and matches the SSPI predictions. The tabular experiments demonstrate a practical application of looping pseudo‑labeling.
> > - By relating multi‑layer attention to iterative pseudo‑labeling and EM, the paper theoretically connects ICL and classical semi‑supervised learning, explaining why depth matters.
> >
> > ### Weaknesses
> > - The theoretical analysis relies on an idealized setting: binary Gaussian mixture with isotropic covariance and task means drawn uniformly from the unit sphere. The assumption of a shared task prior across training and test tasks may not hold in more complex scenarios.
> > - Although the paper shows that deep linear attention can implement polynomial estimators, the constructions rely on hand‑crafted weight matrices rather than learned weights. There is no analysis of whether gradient‑based training can reliably learn these polynomials, nor of how softmax attention or full transformers achieve similar behavior.
> >
> > ### Questions for the authors
> > 1. Your analysis assumes a shared isotropic task prior for both training and test tasks. How sensitive are the results to this assumption? Have you considered tasks drawn from non‑spherical distributions, or mismatched priors between training and inference?
> > 2. In Proposition 1 and Proposition 2, you construct weight matrices to implement polynomial functions of the sample covariance. In practice, are such weights discoverable by gradient‑based training?
> > 3. Can you comment on whether your depth‑induced pseudo‑labeling leads to the kind of “weak‑to‑strong generalization” seen in other semi‑supervised settings, where pseudo‑labels gradually correct errors and expand coverage as more iterations are applied? If so, can the SSPI framework help explain this behavior?

---

> > > ### Author Response · Authors · 2025-08-04
> > >
> > > We thank the reviewer for sharing the comments. Below, we provide a brief response in the interest of time, and we are happy to clarify further if any points remain unclear.
> > >
> > > We refer to our **response to Q1 and Q2** below for **W1 and W2** raised by the reviewer.
> > >
> > >
> > > > *Q1: Your analysis assumes a shared isotropic task prior for both training and test tasks. How sensitive are the results to this assumption? Have you considered tasks drawn from non‑spherical distributions, or mismatched priors between training and inference?*
> > >
> > > **Response:**
> > >
> > > **Non-isotropic distributions:** We first refer the reviewer to our response to **Q2 of Reviewer GrWH**, where we discuss the case of non-isotropic task distributions. In summary, even when the task prior has non-identity covariance (i.e., $\mu \sim \mathcal{N}(0, \Sigma)$), single-layer linear attention still fails to benefit from unlabeled data. In contrast, when the task mean has a nonzero expectation ($\mathbb{E}[\mu] \neq 0$), single-layer models can exploit unlabeled data more effectively. This exploitation relies on incorporating the $x^\top W X^\top X$ component in the prediction. When $\mathbb{E}[\mu] \neq 0$,  $x^\top W X^\top X$ can explain the optimal labeling function $sign(x^\top\mu)$ by predicting the $x^\top\mathbb{E}[\mu]$. We will add a discussion on this in the manuscript.
> > >
> > > **OOD setting:** Next, we also investigated the out-of-distribution (OOD) setting raised by the reviewer. Notably, Eq. (6) in *Theorem 1 characterizes the prediction behavior for an arbitrary input sequence*. Therefore, the prediction rule in Eq. (6) still holds even under distribution shift, and its output remains consistent with the SPI estimator, and SPI estimator still serves as the optimal supervised classifier. Based on it, the classification error depends solely on the norm of $\mu$ (assuming fixed $\sigma$).
> > >
> > > Inspired by the reviewer’s comment, we investigate further under what conditions the distribution of data matters. We find that
> > > - The above conclusion holds only if the noise distribution has identity covariance, i.e.,  $\xi\sim\mathcal{N}(0,\sigma^2 I)$. However, when the covariance of $\xi$ is non-isotropic, the performance is affected (see Table below).
> > > - Additionally, our analysis assumes the model is pretrained on isotropic data. Under such a pretraining process, the optimization of a single-layer attention aligns with the SPI estimator. However, if the model is instead pretrained on data with non-identity covariance, it will no longer converge to the plain SPI estimator, and the covariance of the test distribution may matter  in performance (We did not include additional experiments on it due to time constraints, but we plan to expand the manuscript to incorporate them in the revision.).
> > >
> > > We empirically validate these findings in the table below (with $d=10$). The results show that for the SPI estimator, the covariance of the task mean $\mu$ has no effect on performance, while the covariance of the noise $\xi$ does lead to performance differences.
> > >
> > > | Test Distribution| $n=10$| $n=30$ | $n=50$ |
> > > |-------------------------|-------------------------|-------------------------|-------------------------|
> > > |$\mu\sim\mathcal{N}(0,I)$,$\xi\sim\mathcal{N}(0,I)$| 0.7593 | 0.8082 | 0.8207 |
> > > |$\mu\sim\mathcal{N}(0,\Sigma)$,$\xi\sim\mathcal{N}(0,I)$| 0.7595 | 0.8080 | 0.8206 |
> > > |$\mu\sim\mathcal{N}(0,I)$,$\xi\sim\mathcal{N}(0,\Sigma)$| 0.7362 | 0.7990 | 0.8177 |
> > >
> > >
> > > Beyond the settings we explore, several other generalizations are of interest and deserve further investigation. For example: What is the performance when pretraining on non-isotropic data (which we will add a discussion on)? How does OOD generalization behave for multi-layer attention models? We will incorporate this discussion on OOD and also highlight these future directions.

---

> > > > ### Author Response · Authors · 2025-08-04
> > > >
> > > > > *Q2: In Proposition 1 and Proposition 2, you construct weight matrices to implement polynomial functions of the sample covariance. In practice, are such weights discoverable by gradient‑based training?*
> > > >
> > > > While our work does not focus on the training dynamics of multi-layer models, it introduces novelty by linking the output of multi-layer linear attention to a high-degree polynomial in $(X,y)$ of maximum degree $O(3^L)$. Regardless of the specific output learned by the model, it must be at least as expressive, or likely more complex, than the polynomial structure we characterize. We agree that understanding whether such constructed weights can be learned through gradient-based training is an important and challenging question. Below, we outline why this remains a difficult problem both theoretically and empirically:
> > > > 1. Analyzing the optimization behavior of attention mechanisms, even in the linear case, is highly nontrivial (please refer to our response to **W1 of Reviewer 9Hjq** for more details). Additionally, multi-layer architectures introduce additional levels of interaction and complexity. As a result, characterizing the exact optimization landscape for multi-layer linear attention remains an open challenge. To the best of our knowledge, no prior work has theoretically analyzed this without imposing simplifying assumptions. We view this as an important direction for future exploration.
> > > >
> > > >
> > > > 2. In Proposition 1, we introduce two special cases, label propagation and feature propagation, to provide intuition for how multi-layer attention refines predictions through iteration. Lemma 1 further shows that multi-layer linear attention can implement increasingly expressive functions, represented as a combination of polynomials of different degrees. Ideally, one could compute the optimal coefficients $a_\ell​$ (for $0 \leq \ell \leq (3^L - 3)/2$) in Eq. (9) and compare the resulting estimator with that of an $L$-layer attention model. However, even estimating these coefficients empirically is highly nontrivial due to the size of the function space and the implicit optimization dynamics.
> > > >
> > > >
> > > > 3. Characterizing which weight configuration gradient descent discovers and its relation to our constructions is even more difficult. This would likely require a substantially different and convoluted analysis of how gradient-based training evolves the weights of the multi-layer attention model. We believe this is a fundamental open problem and an exciting direction for future foundational research.
> > > >
> > > >
> > > > While we do not directly show that gradient-based learning recovers the exact constructions in our propositions, our work provides meaningful insight into what multi-layer linear attention is capable of representing and its benefit for semisupervised learning. These (also see **W1 and W3 of Reviewer 9Hjq**) highlight our contribution that goes beyond existing literature.
> > > >
> > > >
> > > >
> > > > > *Q3: Can you comment on whether your depth‑induced pseudo‑labeling leads to the kind of “weak‑to‑strong generalization” seen in other semi‑supervised settings, where pseudo‑labels gradually correct errors and expand coverage as more iterations are applied? If so, can the SSPI framework help explain this behavior?*
> > > >
> > > > Great point. Yes, depth-induced pseudo-labeling in our model mirrors the classic “weak-to-strong” self-training dynamic: each additional layer (or loop) refines the labels produced by the previous stage, progressively correcting residual errors and expanding coverage.
> > > >
> > > > In the multi-layer case, this refinement is transparent, accomplished by constructing higher order polynomials, which allows the model to approximate spectral estimators (degree$\to\infty$).
> > > > In the looping TabPFN setting, multiple passes serve the same purpose: with each iteration the model re-labels the data using its current hypothesis and immediately trains on the improved labels, leading to monotonic gains in test accuracy and a measurable rise in the fraction of correct pseudo-labels.
> > > >
> > > > The SSPI framework formalizes this intuition. Increasing depth (or loop count) raises the “interaction order” term in SSPI, which tightens the approximation-error bound; the theory and our experiments therefore predicts the observed weak-to-strong trajectory.

---

> > > > > ### Comment · Reviewer_HQka · 2025-08-06
> > > > >
> > > > > Thanks for the clarifications and detailed responses. I think this is a valuable and interesting paper that would make a strong contribution to the conference. I’d like to see it accepted.

---

### Official Review · Reviewer_GrWH · 2025-07-03

**Clarity:** 4
**Significance:** 3
**Originality:** 3
**Rating:** 5
**Confidence:** 4

**Summary:**

This paper investigates how transformers can leverage unlabeled data during in-context learning (ICL), a setting the authors term semi-supervised ICL (SS-ICL). The analysis is centered on a binary Gaussian Mixture Model (GMM) where some in-context examples have missing labels.

The core findings are: (1) A  single-layer linear attention model, while optimal in the fully-supervised case, completely fails to use unlabeled data. The optimization landscape leads it to an estimator that only considers labeled examples. (2) Deeper architectures are essential for semi-supervised learning. Multi-layer or looped transformers can utilize unlabeled data by implicitly constructing polynomial estimators. (3) Based on these insights, the authors propose a looping strategy for off-the-shelf tabular foundation models (like TabPFN) to improve their performance on tasks with few labeled examples.

**Questions:**

1. **Role of Softmax and MLPs**: Your experiments show that a full transformer outperforms the linear attention models. What is your hypothesis on the specific roles that softmax attention and MLP layers play in improving semi-supervised performance? Do they allow the model to learn a better polynomial estimator of $X^\top X$ or do they enable a different, more powerful learning mechanism altogether?

2. **Beyond Isotropic Priors**: Your proof for the failure of 1-layer attention hinges on the isotropic task prior, which makes $\mathbb E[X^\top X /n ]$ uninformative. How would your results change for an anisotropic prior (e.g., $\mu \sim \mathcal N(0, \Sigma)$)?  Could a single-layer model then leverage unlabeled data, or would depth still be required?

3. **Connection to Pseudo-Labeling**: You compellingly argue that the label propagation process resembles iterative pseudo-labeling. Could you make this connection more formal?

**Ethical Concerns:**

["NO or VERY MINOR ethics concerns only"]

**Final Justification:**

The paper is good and the rebuttal is clear. I vote for an acceptance.

**Limitations:**

Yes, the authors have adequately addressed the limitations of their work.

**Quality:**

4

**Strengths And Weaknesses:**

### Strength

1. The question of how transformers use weak or unlabeled supervision is highly relevant, especially as context windows grow and high-quality labeled data becomes a bottleneck. This paper provides the first theoretical characterization of this important setting.

2. The paper presents a sharp, insightful contrast between single-layer and multi-layer attention. The proof that single-layer models cannot use unlabeled data in this setting is a strong negative result, which makes the positive result—that depth enables semi-supervised learning—much more compelling.

3. The theoretical insights directly motivate a practical algorithm. The proposed looping strategy for TabPFN is a clever and direct application of the theory that depth/iteration is key.

4. The paper proposes a concrete mechanism for how depth helps: by constructing polynomial functions of the data covariance matrix.

### Weakness

1. The core theoretical results are derived for linear attention models.

2. The analysis relies on an isotropic Gaussian Mixture Model where the task mean is sampled from a uniform sphere. This assumption is crucial for the argument that single-layer attention fails because the expected covariance $\mathbb E[X^\top X /n ]$ becomes uninformative. It's unclear how the results would change under more structured or anisotropic data priors.

---

> ### Author Rebuttal · Authors · 2025-07-31
>
> We thank the reviewer for the insightful comments and suggestions, which we found valuable and have incorporated to improve the clarity and quality of the manuscript.
>
> > *W1: The core theoretical results are derived for linear attention models.*
>
> **Response:** We kindly refer the reviewer to our responses to **W1 and W3 of Reviewer 9Hjq** as well as **our response to Q3** below (where we establish a connection between linear and softmax attention) for a detailed discussion addressing this concern.
>
> > *W2: The analysis relies on an isotropic Gaussian Mixture Model where the task mean is sampled from a uniform sphere. This assumption is crucial for the argument that single-layer attention fails because the expected covariance $\mathbb{E}[X^\top X/n]$ becomes uninformative. It's unclear how the results would change under more structured or anisotropic data priors.*
>
> **Response:** We provide a detailed response and additional experimental results in our **response to Q2** below.
>
> > *Q1: Role of Softmax and MLPs: Your experiments show that a full transformer outperforms the linear attention models. What is your hypothesis on the specific roles that softmax attention and MLP layers play in improving semi-supervised performance? Do they allow the model to learn a better polynomial estimator of $X^\top X$ or do they enable a different, more powerful learning mechanism altogether?*
>
> **Response:** Great question. We believe the improved performance of the full Transformer model comes from mechanisms beyond simply learning higher-order polynomials of $X^\top X$.
> 1. As noted in our Lemma 1, a 5-layer linear attention model can already express polynomial functions of $X^\top X$ up to degree 120. Yet, full Transformers still outperform it (see Fig. 2(c)), suggesting that their advantage is not merely due to increased expressive power over $X^\top X$.
> 2. Attention mechanisms compute output based on pairwise similarities, and given input $X$, they operate as $att(X)=f(X^\top X)$. In contrast, MLPs act directly on the input $MLP(X)=f’(X)$, and therefore do not improve the polynomial estimator of $X^\top X$, but instead provide more flexible and nonlinear feature transformations.
>
> Therefore, we hypothesize that the MLP layers contribute through a more complex, nonlinear data projection mechanism that complements the attention module by capturing feature-level representations beyond pairwise relationships. Understanding the exact theoretical role of MLPs in this context remains an open and promising direction for future work.
>
>
> > *Q2: Beyond Isotropic Priors: Your proof for the failure of 1-layer attention hinges on the isotropic task prior, which makes $\mathbb{E}[X^\top X/n]$ uninformative. How would your results change for an anisotropic prior (e.g., $\mu\sim \mathcal{N}(0,\Sigma))$? Could a single-layer model then leverage unlabeled data, or would depth still be required?*
>
> **Response:** We thank the reviewer for this insightful question. First, when the task prior is changed to $\mu\sim\mathcal{N}(0,\Sigma)$, single-layer linear attention still fails to benefit from unlabeled data. This is because attention computes pairwise similarity as $x_i^\top W x_j$ where $W=W_kW_q^\top$. When $\mu$ has covariance $\Sigma$, the input data satisfies $\mathbb{E}[xx^\top]=\Sigma+\sigma^2 I:=\bar\Sigma$. This can be transformed into the equivalent isotropic case by:
> $$x_i^\top W x_j=\bar x_i \bar\Sigma^{1/2}W\bar\Sigma^{1/2}\bar x_j^\top,$$
> where $\bar x$ has identity (up to a scalar) covariance. Thus, the anisotropic setting effectively reduces to the isotropic case under a linear transformation, and the conclusion that single-layer attention fails to leverage unlabeled data still holds. The first row of the table below supports this where we fix $d=m=10$ and generate non-identity covariance using $\Sigma=A^\top A/trace(A^\top A)$ with a random matrix $A$. Results show that performance remains unchanged with increasing numbers of unlabeled samples.
>
> Motivated by the reviewer’s comment, we further explored when single-layer models can benefit from unlabeled data. From our analysis in the proof of Lemma 2 (Supplementary Material), the prediction from a single-layer model depends on $x^\top X^\top X$ (assuming $W=I$). If $\mathbb{E}[\mu]=0$, then $\mathbb{E}[x^\top X^\top X]=0$, and the unlabeled data does not help. However, if we use a prior with nonzero mean (e.g., $\mu\sim\mathcal{N}(\nu,I)$ where $\nu\neq 0$), the model can extract useful signal from $\mathbb{E}[x^\top X^\top X]\neq 0$. The second row in the table shows that under this setting, increasing unlabeled data improves single-layer model performance: (we set $d=m=10$)
>
> | | $n=10$| $n=50$ | $n=100$ |
> |-------------------------|-------------------------|-------------------------|-------------------------|
> |$\mu\sim\mathcal{N}(0,\Sigma)$| 0.7842 | 0.7842 | 0.7843 |
> |$\mu\sim\mathcal{N}(\nu,I)$| 0.8781 | 0.8928 | 0.8998 |
>
> This is a valuable discussion, and we will incorporate these findings into our revised manuscript.
>
> > *Q3: Connection to Pseudo-Labeling: You compellingly argue that the label propagation process resembles iterative pseudo-labeling. Could you make this connection more formal?*
>
> **Response:** As discussed in Proposition 1 (Label Propagation) and the discussion that follows (Lines 216–221),   multi-layer linear attention can be viewed as an EM-like algorithm, where each layer corresponds to one iteration of pseudo-label refinement. Here, we provide a formal discussion as suggested by the reviewer, and we also refer to our response to **W3 of Reviewer 9Hjq** for further details. We will incorporate it into our revised manuscript.
>
> Specifically, for attention mechanism (subsuming both linear and softmax), at the $\ell$th layer, the pseudo-label assigned to the $i$th example is updated as follows
> $$y_i^\ell\to y_i^\ell+c_\ell\sum_js_{ij}y_j^\ell$$
> where $s_{ij}$ is the weighting strategy that determines how much different tokens influence the token $i$. For linear attention, $s_{ij}=x_i^\top W_\ell x_j$; for softmax attention, $s_{ij}=e^{x_i^\top W_\ell x_j}/\sum_k e^{x_i^\top W_\ell x_k}$. This form resembles the update step in EM/label propagation algorithms (but with different update rules for different models), where the model iteratively updates predictions based on current pseudo-labels.

---

> > ### Comment · Reviewer_GrWH · 2025-08-02
> >
> > Thank you for the clarification and they resolve my questions. I keep my positive review for this paper. Looking forward to the updated manuscript with new discussions. Thanks!

---

### Official Review · Reviewer_9Hjq · 2025-07-03

**Clarity:** 2
**Significance:** 2
**Originality:** 3
**Rating:** 4
**Confidence:** 2

**Summary:**

This paper provides a theoretical proof to understand how linear attention models leverage unlabeled data in in-context learning. The theory is demonstrated under the tabular foundation model setting.

**Questions:**

* Does looping TabPFN-v2 mean that using the output of one model run as input to the next run? I have a concern about the reusage of parameters. I believe this will deteriorate the performance as compared to running the same number but with different layers.

**Ethical Concerns:**

["NO or VERY MINOR ethics concerns only"]

**Final Justification:**

The rebuttal is clear and convincing. It addresses my concerns and questions. Therefore, I have decided to raise my rating.

**Limitations:**

yes

**Quality:**

3

**Strengths And Weaknesses:**

**Strengths**:
* This paper provides a concrete proof of why unlabeled data benefits semi-supervised ICL and points out the key factor of model depth.
* The experiment on tabular models demonstrates the effectiveness of the proposed looping strategy.

**Weaknesses**:
* The analyzed attention module is simplified as w/o softmax version (eq. (3)), differing from mainstream model settings. This may limit the application of the theory.
* The experimented dimension is only set to 10 (Line 285). Why not try a high dimensionality?
* There is a gap between the discussed task setting and the real datasets. For example, it's been widely proven that the attention with softmax outperforms the attention without softmax. However, Figure 2(c) reaches an opposite conclusion. Therefore, the content of this paper may not directly apply to standard Transformer-structured models.

---

> ### Author Rebuttal · Authors · 2025-07-31
>
> We thank the reviewer for recognizing our theoretical contributions. We address the comments below.
>
> > *W1:  The analyzed attention module is simplified as w/o softmax version (eq. (3)), differing from mainstream model settings. This may limit the application of the theory.*
>
> **Response:** In this work, our goal is to analyze the exact optimization behavior of attention mechanisms. Note that 1) attention captures token dependencies through pairwise similarities; and 2) even linear attention induces a non-convex landscape. Therefore, its theoretical analysis is already a challenging and meaningful problem. Prior works (Ahn et al.(2023), Mahankali et al.(2024), Li et al.(2024)) and ours begin with linear attention as a tractable starting point. In contrast, our work considers the semi-supervised setting and examines how model depth affects prediction performance. To the best of our knowledge, this perspective has not been theoretically explored.
>
> While our analysis does not directly cover softmax attention, we empirically compare linear attention, softmax attention, and full Transformers (softmax attention + MLP) in Fig. 2(c). Under our GMM setting, softmax attention without the nonlinear MLP does not outperform linear attention. We provide further discussion and explanation in our **response to W3**, where we describe how **attention mechanisms can generally be formalized as a pseudo-labeling mechanism, applicable to both softmax and linear**. However emphasis is given to linear attention as we can establish concrete and nontrivial theoretical guarantees.
>
> Lastly, although our theoretical results are based on linear attention, they motivate the idea of looping, which we successfully validate in realistic scenarios using real data and TabPFN. These results demonstrate that the theoretical insights carry over and provide practical benefits in more complex settings.
>
> > *W2:  The experimented dimension is only set to 10 (Line 285). Why not try a high dimensionality?*
>
> **Response:** First, since the GMM is a standard and generalizable data model, the dimensionality does not affect our theoretical results. To clearly validate our theoretical findings while keeping the computational cost low, we chose a representative dimension of $d = 10$. Notably, even with such a low dimension, it requires up to 10,000 in-context samples to empirically approach Theorem 2 (see Fig. 1(c), where the green curves approach the black dotted line).
>
> To further address the reviewer's concern, we also conducted additional experiments at higher dimensions (presented in the table). The results are consistent with those in Fig. 1, where we observe that 1-layer=SPI=Theorem 1, and SSPI-1<2-layer<5-layer$\approx$SSPI-$\infty$. (Note: due to time constraints, each setting is evaluated based on a single run, which may introduce variance.)
> |Dimension ($d$)| # labeled ($np$) | # in-context ($n$) | 1-layer| 2-layer | 5-layer | SPI | SSPI-1 | SSPI-$\infty$|Theorem 1
> |-------------------------|-------------------------|-------------------------|:------------:|:------------:|:------------:|:------------:|:------------:|:------------:|:------------:|
> | 50 | 50 | 50 | 0.7602 | 0.7603 | 0.7602 | 0.7602 | 0.7602 | 0.7601 | 0.7602 |
> |  | | 100 | 0.7602 | 0.7696 | 0.7698 | 0.7603 | 0.7668 | 0.7698 |  |
> |  | | 500 | 0.7604 | 0.8167 | 0.8173 | 0.7602 | 0.8129 | 0.8188 |  |
> | 100 | 100 | 100 | 0.7599 | 0.7600 | 0.7601 | 0.7602 | 0.7602 | 0.7601| 0.7602 |
> |  | | 500 | 0.7602 | 0.7979 | 0.7980 | 0.7604 | 0.7954 | 0.7993 |  |
>
> In addition, for the Looping TabPFN experiments, the model employs an embedding dimension of 192.
>
> > *W3: There is a gap between the discussed task setting and the real datasets. For example, it's been widely proven that the attention with softmax outperforms the attention without softmax. However, Figure 2(c) reaches an opposite conclusion. Therefore, the content of this paper may not directly apply to standard Transformer-structured models.*
>
> **Response:**
> We thank the reviewer for raising this important point. While linear and softmax attention differ in their operational functions, they share similarities from a label propagation perspective due to their attention-based structure, and both can be interpreted as a pseudo-labeling algorithm.
>
> A generic attention mechanism with suitable value weights, regardless of linear or softmax, updates the labels via:
> $$y_i^\ell\to y_i^\ell+c_\ell\sum_js_{ij}y_j^\ell.$$
> Here, $s$ is the attention map: $s_{ij}=x_i^\top W_\ell x_j$ for linear attention, and is the softmax similarity between $i$th and $j$th token for softmax attention, both determined by the attention weights at layer $\ell$, where $W_\ell=W_{k\ell}W_{q\ell}^\top$. From this, it can be seen that the first attention layer will impute the missing labels using labeled data based on attention similarities between labeled and unlabeled features. The future layers act as a label propagation mechanism that updates the labels by creating their data-dependent mixtures. We will revise the paper with this exposition to highlight how attention generically implements pseudo-labeling.
>
> We agree that softmax attention is often better than linear attention for **real datasets** but it doesn’t have to be true for **all datasets**. Specifically, for the supervised GMM setting, a single linear attention emulates the Bayes optimal estimator as proven in the paper. This mostly explains why linear attention outperforms softmax attention in Fig 2(c). Although softmax attention underperforms linear attention in isolation, the full Transformer model (softmax attention + MLP) outperforms all other models. (See Fig. 2c).
>
> We also emphasize the following
>
> 1. Linear attention is widely used in theoretical study of transformers (see Ahn et al.(2023), VonOswaldetal.(2023), Mahankali et al.(2024)) and also underlies state-of-the-art architectures like gated-linear attention and Mamba2 (Li et al.(2024), Li et al.(2025)).
> 2. GMM is a classical and established dataset model for the study of classification problems/algorithms.
> 3. The benefit of increased depth for both attention types is evident, supporting our main theoretical claim that depth enhances semi-supervised performance.
> 4. Our theory highlights the utility of looping, a mechanism we validate in real-world settings using TabPFN. These results demonstrate that the insights from our theory translate into practical benefits beyond the simplified setup.
>
> We will include a subsection in our revised manuscript to incorporate this discussion.
>
>
>
>
> > *Q: Does looping TabPFN-v2 mean that using the output of one model run as input to the next run? I have a concern about the reusage of parameters. I believe this will deteriorate the performance as compared to running the same number but with different layers.*
>
> **Response:** Model looping is not a new concept and it has been widely explored and applied in various contexts (see citations below). It offers several advantages: it enables smaller models to handle more complex tasks, allows models to self-correct and refine outputs over multiple passes, and supports adaptability to varying input complexities by adjusting the number of iterations.
>
> The primary benefit of looping is its flexibility and efficiency: In our TabPFN experiments, we did not retrain or redesign a deeper model. Instead, we used the same pretrained model and applied it iteratively in-context. This looping strategy improved performance without requiring additional model capacity or retraining from scratch. At a high-level, an autoregressive LLM can also be viewed as text generation by looping the same parameters. That said, we agree with the reviewer that our results can be enhanced by either (1) training a deeper architecture or (2) optimizing looped TabPFN for semisupervised learning.
>
> - Mohtashami, Amirkeivan, et al. "Cotformer: A chain-of-thought driven architecture with budget-adaptive computation cost at inference." arXiv preprint arXiv:2310.10845 (2023).
>
> - Giannou, Angeliki, et al. "Looped transformers as programmable computers." International Conference on Machine Learning. PMLR, 2023.
>
> - Yu, Qifan, et al. "Enhancing auto-regressive chain-of-thought through loop-aligned reasoning." arXiv preprint arXiv:2502.08482 (2025).

---

> > ### Comment · Reviewer_9Hjq · 2025-08-02
> >
> > Thank you for the detailed clarification. I have no more questions and have raised my rating.

---

### Public Comment · ~Yingcong_Li1 · 2026-01-23

We identified a citation error in the final paragraph of the Related Work section, where LLM-generated incorrect references were included. We have corrected these references to reflect the accurate source and sincerely apologize for the oversight.

---

### Decision · Program_Chairs · 2025-09-17

**Decision:**

Accept (poster)

**Comment:**

The paper provides a theoretical framework explaining how linear attention models leverage unlabeled data in a semi-supervised in-context learning setting. All reviewers were positive, recognizing the paper's theoretical contributions, particularly the insightful contrast between single-layer and multi-layer attention models. The authors' rebuttal successfully addressed all major concerns, including the limitations of the linear attention assumption, the role of deeper architectures, and the effectiveness of the proposed looping strategy on practical datasets like TabPFN. The consensus among reviewers is that the paper makes a strong, clear, and relevant contribution to the field, warranting acceptance.